# On the Empirical Power of Goodness-of-Fit Tests in Watermark Detection

**Weiqing He**[*]
University of Pennsylvania
weiqingh@sas.upenn.edu

**Xiang Li**[*]
University of Pennsylvania
lx10077@upenn.edu

**Tianqi Shang**
University of Pennsylvania
tianqi.shang@pennmedicine.upenn.edu

**Li Shen**[†]
University of Pennsylvania
li.shen@pennmedicine.upenn.edu

**Weijie Su**[†]
University of Pennsylvania
suw@wharton.upenn.edu

**Qi Long**[†]
University of Pennsylvania
qlong@upenn.edu

## Abstract

Large language models (LLMs) raise concerns about content authenticity and integrity because they can generate human-like text at scale. Text watermarks, which embed detectable statistical signals into generated text, offer a provable way to verify content origin. Many detection methods rely on pivotal statistics that are i.i.d. under human-written text, making goodness-of-fit (GoF) tests a natural tool for watermark detection. However, GoF tests remain largely underexplored in this setting. In this paper, we systematically evaluate eight GoF tests across three popular watermarking schemes, using three open-source LLMs, two datasets, various generation temperatures, and multiple post-editing methods. We find that general GoF tests can improve both the detection power and robustness of watermark detectors. Notably, we observe that text repetition, common in low-temperature settings, gives GoF tests a unique advantage not exploited by existing methods. Our results highlight that classic GoF tests are a simple yet powerful and underused tool for watermark detection in LLMs.[3]

## 1 Introduction

The rapid advancement of large language models (LLMs) has enabled machines to generate fluent and coherent text that closely resembles human writing. While this progress has unlocked many applications, it has also raised significant concerns about the authenticity, ownership, and integrity of generated content. For example, LLMs can be misused to spread misinformation [58, 55, 51], fabricate academic work [52, 38], or manipulate digital communication at scale [45, 49, 8]. These risks have prompted growing interest in developing reliable tools to trace and verify machine-generated text.

A promising solution is text watermarking, which embeds hidden statistical signals into generated text. Specically, it introduces dependencies between tokens $w_t$ and secret pseudorandom numbers $\zeta_t$ [5, 1]. In watermarked generation, each token $w_t \sim \boldsymbol{P}_t$ is produced by a decoding function $\mathcal{S}$ as $w_t = \mathcal{S}(\boldsymbol{P}_t, \zeta_t)$, where $\boldsymbol{P}_t$ is the next-token predictive (NTP) distribution that $w_t$ follows. This

---

[*]Equal contribution.
[†]Corresponding authors.
[3]Code is available at https://github.com/hwq0726/GoF-for-Watermark-Detection.

39th Conference on Neural Information Processing Systems (NeurIPS 2025).

induces a watermark signal—statistical dependence between $w_t$ and $\zeta_t$. In contrast, human-written text lacks this dependence, as humans do not have access to $\zeta_t$. Detection methods leverage this difference using scalar pivotal statistics [31], defined as $Y_t = Y(w_t, \zeta_t)$, within a hypothesis testing framework:

$$H_0 : w_{1:n} \text{ is human-written} \quad \text{vs.} \quad H_1 : w_{1:n} \text{ is LLM-generated.} \quad (1)$$

A key property of the pivotal function $Y$ is that $Y(w, \zeta)$ follows a known distribution $\mu_0$ whenever $w$ is independent of $\zeta$, which exactly holds for human-written $w$ since human users do not know the secret $\zeta$. Thus, problem (1) can be reformulated in terms of $Y_t$:

$$H_0 : Y_t \sim \mu_0 \text{ i.i.d.} \quad \text{vs.} \quad H_1 : Y_t \sim \text{other distribution that depends on } \boldsymbol{P}_t. \quad (2)$$

The formulation in (2) is standard and popular in the watermarking literature [31, 29, 9, 28, 1, 16, 5, 14, 60, 61, 27, 21, 56]. Many detection rules, though somewhat ad hoc, are designed to address this problem. In the idealized setting where all NTP distributions $\{\boldsymbol{P}_t\}_{t=1}^n$ are known, the log-likelihood ratio test based on the pivotal statistics $\{Y_t\}_{t=1}^n$ is provably optimal [22]. In practice, however, these distributions are often unavailable due to limited access to commercial LLMs and unknown prompts. To overcome this, recent works have developed NTP-agnostic approaches. For example, Li et al. [31] assumes all $\boldsymbol{P}_t$ lie within a known class and derives the least favorable detection rule using a minimax formulation. This framework is extended by Li et al. [33] to handle human editing, where the observed text may contain a mix of human-written and LLM-generated content. In this case, the alternative hypothesis in (2) is modified to reflect the fraction of human-edited tokens. Nonetheless, both works share the same null hypothesis $H_0$: $\{Y_t\}_{t=1}^n$ are i.i.d. samples from the known $\mu_0$.

This structure reveals a close connection between watermark detection and the classical statistical problem of goodness-of-fit (GoF) testing. GoF tests aim to determine whether i.i.d. samples follow a given distribution. While watermark detection involves non-i.i.d. data under $H_1$—due to the autoregressive nature of LLMs, where each $\boldsymbol{P}_t$ depends on the history $w_{1:(t-1)}$—GoF tests remain applicable because $H_0$ is the same: $\{Y_t\}_{t=1}^n$ are i.i.d. from a known $\mu_0$. Despite their long-standing role in statistics [7], *GoF methods have received limited attention in the watermarking literature*. Prior work has largely focused on designing new watermarking schemes, with less emphasis on improving detection power (see Appendix B for a detailed discussion). Li et al. [33] proposed a truncated divergence-based GoF test and analyzed its theoretical performance on the Gumbel-max watermark [1], but its effectiveness on other watermarking schemes remains unclear, as does the performance of other classical GoF tests. This raises a natural question: *how well do classical GoF tests work for modern watermark detection?* A systematic study of classical GoF tests could provide valuable insights for developing robust and effective watermark detection methods.

**Our contribution.** In this work, we study this connection empirically, and our contributions are:

- **Systematic empirical evaluation.** We introduce general GoF tests to watermark detection and systematically evaluate eight GoF detection rules across three popular watermarking schemes. Our experiments span diverse configurations, including three open-source language models, two datasets, four temperature settings, and three types of edits.

- **Understanding GoF advantages.** GoF tests consistently outperform baseline methods across varying temperatures and text lengths, and this robustness stems from their ability to capture distribution-level deviations in the pivotal statistics. At high temperatures, watermark signals are stronger due to increased entropy in the next-token distributions. This leads to more noticeable shifts between the empirical CDF and the null CDF, which GoF tests are particularly effective at detecting—even with moderate text lengths. At low temperatures, watermark signals tend to weaken, making detection more challenging. However, lower temperatures also induce repetition in the generated text, introducing structured patterns that GoF tests can exploit. This repetition results in deviations from the expected null CDF, enabling GoF tests to maintain strong detection power even in low-entropy settings. These complementary advantages explain why GoF tests are effective across a wide range of generation scenarios, including those that are traditionally hard for watermark detection.

- **Robustness to modification.** We evaluate GoF tests under both common text edits (deletion and substitution) and information-rich edits. In all cases, GoF-based methods maintain high detection power, demonstrating strong robustness to modifications.

In short, our results show that GoF tests are a valuable complement to existing watermark detection methods, improving both detection power and robustness across a wide range of conditions. Due to space constraints, we defer discussion of related work to Appendix B.

## 2 Preliminaries

**Watermark embedding.** LLMs like the GPT series [44, 4] generate text autoregressively, producing one token at a time based on previous tokens. Each token is drawn from a fixed vocabulary $\mathcal{W}$ using a multinomial distribution $\boldsymbol{P}_t = (P_{t,w})_{w \in \mathcal{W}}$, called the next-token prediction (NTP) distribution. This distribution depends on the previous tokens, user prompts, and system-level prompts, summarized as $\boldsymbol{P}_t = \mathcal{M}(w_{1:(t-1)})$ [53, 44, 4]. To embed a watermark, the model uses a pseudorandom variable $\zeta_t$, computed by a deterministic hash function: $\zeta_t = \mathcal{A}(w_{(t-m):(t-1)}, \texttt{Key})$, where $\mathcal{A}$ is a hash function, $m$ is a context window size, and $\texttt{Key}$ is a secret key. Once $\zeta_t$ is computed, the next token is generated as $w_t = \mathcal{S}(\boldsymbol{P}_t, \zeta_t)$, where $\mathcal{S}$ is a (possibly stochastic) decoder. The decoder $\mathcal{S}$ is said to be unbiased if for any $\boldsymbol{P}$ and $w$, it satisfies $\mathbb{P}_{\zeta \sim \pi}(\mathcal{S}(\boldsymbol{P}, \zeta) = w) = P_w$. This means that when $\zeta$ is truly random, $\mathcal{S}(\boldsymbol{P}, \zeta)$ still follows the original NTP distribution $\boldsymbol{P}$. In other words, an unbiased decoder samples from the intended NTP.

**Watermark detection.** As introduced in Section 1, watermark detection aims to identify statistical dependence between each token $w_t$ and the pseudorandom number $\zeta_t$. This is done using pivotal statistics, leading to the hypothesis testing problem in (2) [31]. Under $H_0$, there is no watermark, and $w_t$ and $\zeta_t$ are assumed to be independent. By design, the pivotal statistic $Y_t = Y(w_t, \zeta_t)$ then follows a known distribution $\mu_0$, regardless of how $w_t$ is distributed. Under the alternative, the watermarking process introduces dependence through the decoder $\mathcal{S}$, causing $Y_t$ to deviate from $\mu_0$. In particular, $Y_t \mid \boldsymbol{P}_t$ follows a distribution $\mu_{1,\boldsymbol{P}_t}$ that depends only on $\boldsymbol{P}_t$. This leads to the following hypothesis testing problem:

$$H_0 : Y_t \sim \mu_0 \text{ i.i.d.} \quad \text{vs.} \quad H_1 : Y_t \mid \boldsymbol{P}_t \sim \mu_{1,\boldsymbol{P}_t}. \tag{3}$$

Li et al. [33, 32] extend it to a more practical setting where the generated text may be partially edited. They model human edits as a mixture effect so that $Y_t$ is from a mixture of $\mu_0$ and $\mu_{1,\boldsymbol{P}_t}$ under $H_1$.

**Three considered unbiased watermarks.** We consider three unbiased watermarking schemes. For brevity, we describe only their null distributions $\mu_0$, which suffice for understanding our methods and results. Full details are provided in Appendix A. The first is the Gumbel-max watermark [1], where $\mu_0$ is the uniform distribution $U(0, 1)$. The second is the inverse transform watermark [29], whose null CDF satisfies $\mu_0(Y \leq r) = r^2$ for $r \in [0, 1]$. The third is Google's SynthID watermark [9], where the pivotal statistic under $H_0$ follows the distribution of $\frac{1}{k}\text{IrwinHall}(k)$ where $\text{IrwinHall}(k)$ is the Irwin–Hall distribution, i.e., the sum of $k$ i.i.d. $U(0, 1)$ variables. A common feature of all three is that their null distributions $\mu_0$ are known and have computable CDFs.

**An exception.** The green-red list watermark [27, 28, 60] (see Appendix A.4) is a popular method. It randomly partitions the vocabulary into a green (favored) and red (disfavored) set, boosts the probabilities of green tokens, and samples from the perturbed NTP distribution. Detection is based on counting green tokens and rejecting $H_0$ if their frequency exceeds a threshold. Variants [56, 21] use more carefully designed partitions but follow the same overall pipeline. We exclude these methods from our experiments because GoF tests reduce to the original detection rule in this setting. The pivotal statistic is binary (whether a token is green), and the order of them does not matter—only the total count of green tokens is informative. As a result, both methods are effectively equivalent, making additional experiments unnecessary.

## 3 GoF Tests for Watermark Detection

**Rationale for using GoF tests.** Goodness-of-fit (GoF) tests are classical tools in statistics for determining whether a sequence of i.i.d. samples comes from a specified distribution [7]. The standard formulation is:

$$H_0 : Y_t \sim \mu_0 \text{ i.i.d. } \forall t \quad \text{vs.} \quad H_1^{\text{standard}} : Y_t \sim \mu_1 \text{ i.i.d. } \forall t, \tag{4}$$

where $\mu_1 \neq \mu_0$. GoF tests evaluate deviations from the null distribution $\mu_0$ and determine whether they are statistically significant. However, this classical setup does not directly apply to watermark detection, where the alternative hypothesis $H_1$ in (3) involves structured dependence introduced by the watermarking process. Specifically, the pivotal statistics $\{Y_t\}_{t=1}^n$ are not i.i.d. under $H_1$ due to the autoregressive nature of LLMs and the time-varying NTP distributions $\boldsymbol{P}_t$.

However, the core principle of GoF tests—measuring deviation from the null distribution $\mu_0$—remains highly relevant. The most related work, Li et al. [33], proposed a divergence-based GoF test [24]

Table 1: Considered GoF tests.

| Tests name | Deviation measure | Tests name | Deviation measure |
|---|---|---|---|
| Tr-GoF test, Phi, [33] | $S_n^+(s) = \sup\limits_{r \in [\mathsf{p}^+, 1)} K_s^+(F_n(r), F_0(r))$ | Kuiper's test, Kui, [30] | $V_n = \max\limits_{1 \le i \le n} \left( \frac{i}{n} - \mathsf{p}_{(i)} \right) + \max\limits_{1 \le i \le n} \left( \mathsf{p}_{(i)} - \frac{i-1}{n} \right)$ |
| Kolmogorov-Smirnov test Kol, [50] | $D_n = \max\limits_{1 \le i \le n} \left\{ \max \left( \mathsf{p}_{(i)} - \frac{i-1}{n}, \frac{i}{n} - \mathsf{p}_{(i)} \right) \right\}$ | Anderson-Darling test, And, [2] | $A^2 = -n - \frac{1}{n} \sum_{i=1}^n (2i-1) \log \big[ (1 - \mathsf{p}_{(n+1-i)}) \mathsf{p}_{(i)} \big]$ |
| Cramér-von Mises test, Cra, [6] | $W^2 = \frac{1}{12n} + \sum_{i=1}^n \left( \mathsf{p}_{(i)} - \frac{2i-1}{2n} \right)^2$ | Watson's test, Wat, [54] | $U^2 = W^2 - n \left( \bar{F} - \frac{1}{2} \right)^2$ |
| Neyman's smooth test, Ney, [40] | $T_k = n \sum_{j=1}^k a_j^2$ | Chi-squared test, Chi, [42] | $\chi^2 = \sum_{i=1}^k \frac{(O_i - E_i)^2}{E_i}$ |

for detecting Gumbel-max watermarks and demonstrated its asymptotic robustness. Yet, they did not establish its uniqueness or compare it against other GoF tests. Moreover, their analysis relies on asymptotic assumptions such as infinite text length, while in practice, the finite-sample performance of GoF tests can vary substantially—even among asymptotically optimal methods. Our work complements this line of research by empirically evaluating classical GoF tests in the context of watermark detection. We also investigate which factors most affect their performance.

**Introduction of GoF tests.** All GoF tests evaluate deviations from the null distribution $\mu_0$ and reject $H_0$ if the deviation exceeds a specified threshold. Thus, it suffices to define the measures of deviation used by each test. Before doing so, we introduce some notation. Let $Y_1, \ldots, Y_n$ denote the pivotal statistics, and let $Y_{(1)} \le Y_{(2)} \le \cdots \le Y_{(n)}$ be their order statistics. The empirical cumulative distribution function (CDF) is defined as $F_n(r) = \frac{1}{n} \sum_{t=1}^n \mathbf{1}_{Y_t \le r}$, and the theoretical CDF under the null is $F_0(r) = \mu_0(Y \le r)$. Recall that the null distribution $\mu_0$ is known but varies by watermarking scheme (see Section 2). Given this, we can also compute $p$-values as $\mathsf{p}_t = 1 - F_0(Y_t)$.

Eight GoF tests are summarized in Table 1, with each referred to by a three-letter abbreviation (e.g., the Kolmogorov-Smirnov test as Kol). All GoF tests operate on pivotal statistics and are permutation-invariant. Therefore, the order of pivotal statistics does not affect their detection results. Below, we provide additional details omitted from the table for brevity:

- Tr-GoF test [33] uses a truncated $\phi$-divergence [24], which is the reason for its name Phi. The truncation is introduced via $c_n^+$ satisfying $\mathsf{p}^+ = \sup\{\mathsf{p}_{(t)} : \mathsf{p}_{(t)} \le c_n^+\}$ to ensure numerical stability. Here, $K_s^+(u, v) = K_s(u, v) \mathbf{1}\{0 < v < u < 1\}$ and $K_s(u, v) = v \phi_s \left( \frac{u}{v} \right) + (1 - v) \phi_s \left( \frac{1-u}{1-v} \right)$ is the untruncated $\phi_s$-diveregence. The function $\phi_s(x)$ is convex and defined as:

$$\phi_s(x) = \begin{cases} x \log x - x + 1, & \text{if } s = 1, \\ \frac{1 - s + sx - x^s}{s(1-s)}, & \text{if } s \ne 0, 1, \\ -\log x + x - 1, & \text{if } s = 0. \end{cases}$$

  In our experiment, we fix $s = 2$ for all considered watermarks due to its good performance [33].

- Kolmogorov-Smirnov test (Kol) measures the largest difference between the empirical and null CDFs. In fact, one can show that $D_n = \sup_x |F_n(x) - F_0(x)|$.

- Watson's test adjusts the Cramér-von Mises statistic $W^2$ to ensure invariance to location changes, so $U^2 = W^2 - n \left( \bar{F} - \frac{1}{2} \right)^2$ where $\bar{F} = \frac{1}{n} \sum_{i=1}^n F(Y_{(i)})$.

- Neyman's smooth test (Ney) expands deviations from the null distribution using orthonormal polynomials. Specifically, it computes coefficients

$$a_j = \frac{1}{\sqrt{\lambda_j}} \sum_{i=1}^n h_j (Y_i),$$

  where $h_j$ are orthonormal Legendre polynomials for $U(0, 1)$ and $\lambda_j$ are normalizing constants. The test statistic then aggregates the first $k$ coefficients as $T_k = n \sum_{j=1}^k a_j^2$, with $k = 3$ used in our experiments.

- Chi-squared test (Chi) divides the range of $\mu_0$ into $k$ equal-width bins, counts observed frequencies $O_i$, and computes expected frequencies $E_i = n/k$. The test statistic is $\chi^2 = \sum_{i=1}^k \frac{(O_i - E_i)^2}{E_i}$ with critical values drawn from the chi-squared distribution with $k - 1$ degrees of freedom [42].

Importantly, for most GoF tests, the exact null distribution does not admit a closed-form expression; instead, only an asymptotic distribution emerges as the token length $n$ grows. Nevertheless, our

---

**Algorithm 1** GoF test for watermark detection (example: `Kol` for the Gumbel–max watermark)

---

**Require:** Token sequence $w_{1:n}$; watermark decoder $\mathcal{S}$; significance level $\alpha$; CDF under the null $F_0$.
1: **Compute pivotal statistics** $Y_1, \ldots, Y_n$ from the sequence $w_{1:n}$.
2: **Compute $p$-values:** $\mathrm{p}_t = 1 - F_0(Y_t)$, $t = 1, \ldots, n$.
3: **Sort** the $p$-values in ascending order: $\mathrm{p}_{(1)} \leq \cdots \leq \mathrm{p}_{(n)}$.
4: **Compute the test statistic** (`Kol`)
$$D_n \leftarrow \max_{1 \leq i \leq n} \max\left( \mathrm{p}_{(i)} - \tfrac{i-1}{n}, \ \tfrac{i}{n} - \mathrm{p}_{(i)} \right).$$
5: **Estimate critical value** $\gamma_\alpha$ based on the information of the watermarking scheme.
6: **if** $D_n > \gamma_\alpha$ **then**
7:      **Reject** $H_0$
8: **else**
9:      **Do not reject** $H_0$
10: **end if**

---

experiments show that relying on this large-sample approximation still yields reliable Type I error control in practice. We defer a detailed discussion of their practical considerations to Appendix C. To conclude, we integrate the full detection procedure into Algorithm 1, providing a streamlined view of how the Kolmogorov–Smirnov test (`Kol`) can be applied to the Gumbel-max watermark. The same procedure can be straightforwardly applied to other GoF tests, which readers may extend as needed.

## 4 Language Model Experiments

### 4.1 Experiment settings

**Experimental setup** In our evaluation, we consider three open-source LLMs—OPT-1.3B, OPT-13B [59], and Llama 3.1-8B [11]—across four temperature settings: $\mathrm{T} \in \{0.1, 0.3, 0.7, 1.0\}$. We evaluate watermark performance on two text generation tasks: (i) text completion and (ii) long-form question answering. For text completion, we use the `C4` dataset [46]. Each document in the dataset is truncated to the first 50 tokens, which serve as prompts for the LLM to complete. For long-form question answering, we use the `ELI5` dataset [12], where the LLM generates detailed answers to given questions. In both tasks, we randomly sample 1,000 documents and conduct experiments using three LLMs across four temperature settings, following a consistent pipeline. As the relative performance of different GoF tests is similar between tasks, we present the results on text completion in this section and defer the detailed results for the `ELI5` dataset to Appendix D.4.

**Remark 4.1** (Why we don't include green-red list watermark). *As discussed in Section 2, we do not include the green-red list watermark [27] or its variants in our experiments. This is because their pivotal statistics are binary (indicating whether a token is green), and the order of these values doesn't play a role. As a result, the original detection rule—counting the number of green tokens—is already effective, and GoF tests reduce to this same procedure. Therefore, additional experiments are unnecessary.*

**Common text edits.** Human edits can weaken watermark signals [48, 19, 33]. To evaluate detection robustness, we apply two common types of edits: *word deletion* and *synonym substitution*. For each, we randomly select a fraction $r_{\mathrm{edit}} \in \{0.1, 0.2\}$ of watermarked tokens and either delete them or replace them with synonyms from WordNet [39].

**Information-rich edits.** We consider a stronger editing setting in which the hash function $\mathcal{A}$ and secret key `Key` are known to the user. In this case, the user can selectively modify a limited number of LLM-generated tokens to reduce watermark signals while preserving the overall quality of the text. We refer to this targeted modification as an *information-rich edit*. With a limited token budget, the optimal strategy is to change tokens carrying the strongest watermark signals. In all three considered schemes, tokens with higher pivotal statistics tend to have stronger signals. Thus, replacing these tokens lowers the overall signal strength significantly. To simulate this, we compute the pivotal statistics for all tokens in a watermarked sequence. Then, we select a fraction $r_{\mathrm{info}} \in \{0.3, 0.5\}$ of tokens with the highest statistics and overwrite their values with samples drawn from the null distribution $\mu_0$.

Table 2: Type I errors on human data and Type II errors (averaged over three LLMs) on the C4 dataset. All values are enlarged by 100 for readability. T denotes temperature and $n$ the token length. Baseline refers to the best-performing method among the baseline detectors. Red shading highlights lower values; blue indicates higher values.

| | T | $n$ | Baseline | Phi | Kui | Kol | And | Cra | Wat | Ney | Chi |
|---|---|---|---|---|---|---|---|---|---|---|---|
| **Gumbel-max** | 0.3 | 200 | 18.5 | 21.0 | 26.3 | 19.5 | **15.5** | 21.2 | 36.8 | 19.7 | 18.5 |
| | | 400 | 15.1 | 5.7 | 4.7 | 4.7 | 4.9 | 8.4 | 10.7 | 8.0 | **2.9** |
| | 0.7 | 200 | 0.6 | **0.3** | 0.5 | 0.6 | 0.5 | 0.7 | 0.9 | 0.5 | **0.3** |
| | | 400 | 0.7 | **0.2** | **0.2** | 0.3 | **0.2** | 0.4 | 0.4 | **0.2** | **0.2** |
| | Type I | | - | 0.4 | 0.9 | 1.5 | 0.6 | 0.7 | 1.2 | 1.1 | 0.9 |
| **Inverse-tran.** | 0.3 | 200 | 38.7 | 51.0 | 29.2 | 29.7 | 33.6 | 36.7 | 40.4 | 37.3 | **21.8** |
| | | 400 | 27.1 | 12.1 | 6.0 | 7.4 | 9.3 | 14.0 | 10.7 | 13.0 | **3.6** |
| | 0.7 | 200 | 1.3 | 2.7 | 1.3 | **0.9** | **0.9** | 1.0 | 1.9 | 1.2 | 2.3 |
| | | 400 | 1.5 | 0.5 | **0.2** | 0.3 | 0.4 | 0.6 | 0.4 | 0.5 | **0.2** |
| | Type I | | - | 0.4 | 1.4 | 1.2 | 1.0 | 1.0 | 1.4 | 1.5 | 0.6 |
| **SynthID** | 0.3 | 200 | 58.8 | 61.6 | 49.8 | 53.0 | 53.5 | 57.2 | 61.3 | 57.4 | **36.4** |
| | | 400 | 44.4 | 25.0 | 16.7 | 21.0 | 24.5 | 31.5 | 26.5 | 29.1 | **10.4** |
| | 0.7 | 200 | 2.8 | 3.4 | 4.5 | 2.8 | **2.3** | 3.3 | 8.4 | 3.0 | 3.3 |
| | | 400 | 2.2 | 0.7 | 1.2 | 0.8 | **0.6** | 1.4 | 1.8 | 1.3 | 0.8 |
| | Type I | | - | 0.9 | 1.2 | 0.9 | 1.1 | 1.0 | 1.4 | 1.0 | 1.2 |

**Remark 4.2** (Differences from watermark stealing). *Information-rich edits are related to watermark stealing [26] in that they simulate post-stealing detection, but they are conceptually distinct. The former assumes the watermark information (e.g., secret key) has already been compromised, and the user is allowed a limited editing budget. In contrast, watermark stealing typically focuses on extracting the watermark itself or defending against such attacks [47]. Our study, by contrast, focuses on the detection stage—evaluating how detection rules perform under an idealized attack scenario.*

**Five baselines.** Previous work primarily relies on sum-based detection rules, which reject $H_0$ if the sum $\sum_{t=1}^{n} h(Y_t)$ exceeds a critical threshold, where $h$ is a predefined score function. We adopt five score functions from prior studies as baselines. For the Gumbel-max watermark, we use Aaronson's score [1] and the log score [29, 13]. For the Inverse transform watermark, we apply the negative-sum score [29]. For the SynthID watermark, we use the identity-sum score [9]. In addition, we apply the least-favorable test from Li et al. [31] to all three schemes. For each watermark, we report the best-performing baseline among these five applicable rules. Detailed definitions of the score functions are provided in Table 5 in the Appendix.

### 4.2 Results on Statistical Power

Building on the experimental setup introduced earlier, we now present our findings. In watermark detection, a Type I error corresponds to falsely identifying non-watermarked text as watermarked, while a Type II error arises when a true watermark goes undetected. We evaluate the statistical power of the considered GoF tests by analyzing both error types. To control the Type I error at $\alpha = 0.01$, we adjust the critical values using either theoretical distributions or Monte Carlo simulations. We then examine whether the Type I error is well-controlled and explore how the Type II error decreases under different configurations.

**Type I error control.** We begin by evaluating Type I error control. To this end, we randomly sample 1,000 human-written texts from the C4 dataset and assess Type I error at a significance level of $\alpha = 0.01$, a standard choice in prior work [9, 33, 29, 31]. Table 2 reports the empirical Type I errors at a sequence length of 400 tokens. All detection methods maintain errors close to the target level of 0.01, indicating that Type I error can be effectively controlled for most GoF tests.

**Type II error decay.** We next evaluate Type II errors at the fixed significance level $\alpha = 0.01$. Table 2 reports the average Type II errors across three LLMs for various detection methods. We consider both low temperature (T = 0.3) and high temperature (T = 0.7) settings, using the C4

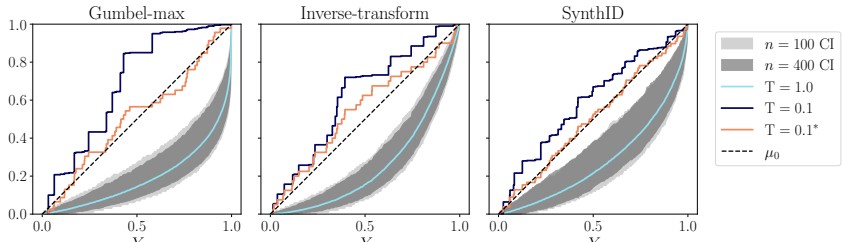

Figure 1: Empirical CDFs of pivotal statistics ($F_0(Y_t)$) under different temperatures T and text lengths $n$. Shaded regions show 95% confidence intervals (CI): light gray for $n = 100$ and dark gray for $n = 400$, both at T = 1.0. At T = 0.1, the dark blue curve shows the empirical CDF of raw scores, while the orange curve (labeled T = $0.1^*$) shows the CDF after removing repeated values.

dataset. Additional results on the `ELI5` dataset are presented in Table 10 in the Appendix. Detailed breakdowns by LLM and task are provided in Appendix D.4 for completeness. We observe from Table 2 that GoF tests generally outperform the baselines across different temperatures and text lengths. This observation can be refined into two points:

1. *Advantages are more pronounced at higher temperature settings.* GoF tests consistently outperform baseline methods at high temperatures, regardless of text length. For instance, under the Gumbel-max watermark at temperature T = 0.7, the baseline yields a Type II error of approximately 0.7% for both $n = 200$ and $n = 400$, whereas several GoF tests achieve lower error rates around 0.3%. This pattern remains consistent across different watermarking schemes.

2. *Advantages persist under low-temperature settings.* Surprisingly, GoF tests maintain a strong advantage even when the temperature is low. For example, under the Gumbel-max watermark with T = 0.3 and $n = 400$, the best baseline yields a 15.1% Type II error, while `And` achieves 4.9% and `Chi` achieves the best 2.9%. This pattern remains consistent for shorter lengths and different watermarking schemes.

**Why do GoF tests perform well at high temperatures.**   This observation naturally leads to the question: Why do GoF tests perform so well across varying temperatures and text lengths? The key reason lies in how they utilize the full empirical distribution (CDF) of the pivotal statistics to detect deviations from the null $\mu_0$. As discussed in Section 3, GoF tests compare the empirical CDF $F_n$ with the known null CDF $F_0$, often using richer nonlinear operations—such as sorting, maximization, or transformations—that allow them to capture subtle discrepancies more effectively. In contrast, baseline methods typically rely on sum-based statistics, which compress the data into a single value. This approach often overlooks structural differences in the distribution and is less sensitive to nuanced deviations.

These properties make GoF tests particularly effective at high temperatures, where watermark signals are strong. In this setting, the empirical CDF diverges noticeably from the null CDF, even for moderate sequence lengths. As shown in Figure 1, the difference is already apparent at $n = 100$, and increasing $n$ further primarily reduces statistical variance. This implies that, under high temperatures, GoF performance becomes less dependent on text length—explaining their consistent advantage across both short and long sequences.

**A missing factor at low temperatures.**   Low-temperature settings are often considered difficult for watermark detection due to weaker signals. Yet, they are common in applications such as customer service, factual QA, and code generation. At lower temperatures, LLM outputs become more deterministic, reducing entropy in both the generated text and the corresponding NTP distributions. This lower variability makes detection harder, as the null and alternative distributions—$\mu_0$ and $\mu_{1,P_t}$—become nearly indistinguishable. Prior work [31] even shows that, in some cases, accurate detection may be information-theoretically impossible. However, our study identifies an overlooked factor: text repetition, which becomes more prominent at low temperatures.

To quantify this, we measure the $m$-gram repetition rate—the proportion of repeated text segments $w_{(t-m):(t-1)}$ across generated outputs. We evaluate this metric under various temperature settings (T $\in \{0.1, 0.3, 0.7, 1.0\}$) using three LLMs with watermarked outputs. For comparison, we also compute the same metric on human-written texts sampled from the `C4` dataset. Figure 2 shows the results for the Gumbel-max watermark; additional results for other watermark schemes appear in

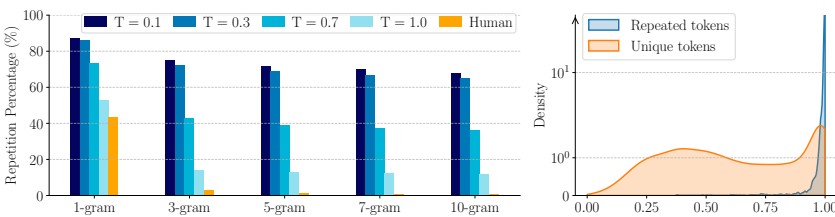

Figure 2: **Left:** Average repetition rate in Gumbel-max watermarked texts across three LLMs, compared to human-written text. **Right:** Distribution of the highest probability in NTP distributions for OPT-1.3B using the Gumbel-max watermark at temperature 0.7.

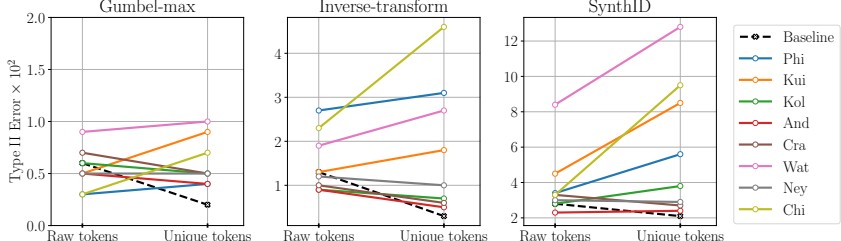

Figure 3: Change in Type II error after removing repeated pivotal statistics (averaged over three LLMs). *Raw tokens* denotes the error rate on the original 200-token sequence; *Unique tokens* denotes the error rate after excluding repeated pivotal statistics from the same sequence.

Appendix D.4 (Figure 8,9). Our findings confirm that *repetition is common in LLM-generated text*, consistent with prior studies[20, 15, 57], and that *lower temperatures tend to amplify this effect*.

**The influence of repetition on top probabilities.** By the hash rule $\zeta_t = \mathcal{A}(w_{(t-m):(t-1)}, \text{Key})$, repeated $m$-grams (i.e., $w_{(t-m):(t-1)} = w_{(t'-m):(t'-1)}$ for some $t' \neq t$) lead to identical pseudo-random values, $\zeta_t = \zeta_{t'}$. According to the decoder rule $w_t = \mathcal{S}(\boldsymbol{P}_t, \zeta_t)$, this does not guarantee $w_t = w_{t'}$ unless $\boldsymbol{P}_t \approx \boldsymbol{P}_{t'}$. This suggests that NTP distributions vary little across repeated segments, leading to repeated token outputs. To examine this, we analyze the top probability (i.e., $\max_w P_{t,w}$) of generated tokens from the OPT-1.3B model with the Gumbel-max watermark at temperature 0.7. As shown in Figure 2 (see Figures 8, 9 for other watermarks), repeated tokens show a sharp spike at 1, indicating near-deterministic generation. In contrast, unique tokens exhibit a more spread-out distribution, reflecting greater diversity. These results suggest that *repetition reinforces low-entropy behavior*.

**The influence of repetition on the empirical CDF.** Repetition also affects the empirical CDF of the pivotal statistics $Y_t$. In ideal low-temperature settings, $\mu_{1,\boldsymbol{P}}$ is information-theoretically close to $\mu_0$ due to low entropy, so the empirical CDF of $Y_t$ should resemble $\mu_0$. However, repetition disrupts this, causing $Y_t$ to deviate further from $\mu_0$. To illustrate, we analyze a sentence generated by OPT-1.3B and plot the empirical CDFs at temperatures 0.1 and 1.0 (Figure 1). At both temperatures, the empirical CDF deviates from $\mu_0$, but in opposite directions. Intuitively, *repetition introduces artificial structure*—evident in the step-like shape of the CDF at low temperature—*that pushes the distribution away from the null*. Removing repeated values smooths the CDF and brings it closer to $\mu_0$. This explains why removing repeated pivotal statistics slightly degrades the performance of GoF tests (see Figure 3). Nonetheless, their performance remains comparable to the baselines.

**GoF tests are effective at capturing CDF differences.** We now summarize why GoF tests perform so well: *they are particularly effective at capturing differences between the empirical CDF and the null $\mu_0$*. At high temperatures, strong watermark signals push the empirical CDF away from $\mu_0$. At low temperatures, repetition introduces additional structure that subtly distorts the CDF. While such deviations may be hard to detect using simple sum-based baselines, GoF tests are specifically designed to identify such distributional shifts—whether large or subtle. This ability to exploit CDF-level differences explains their consistent advantage across watermarking schemes, temperatures, and editing settings. These findings suggest that GoF-based detection is a promising and broadly applicable direction for future watermark detection research.

Table 3: Type II errors (averaged over three LLMs) on the `C4` dataset with temperature 1.0 under various editing types. "Del" denotes word deletion, "Sub" represents synonym substitution, and "Info" refers to information-rich edits. All values are enlarged by 100 for readability. `Baseline` refers to the best-performing method among the baseline detectors. Red shading highlights lower values; blue indicates higher values.

| | Edits | | Baseline | Phi | Kui | Kol | And | Cra | Wat | Ney | Chi |
|---|---|---|---|---|---|---|---|---|---|---|---|
| **Gumbel-max** | Del | @ 0.1 | 0.4 | 0.3 | 0.5 | 0.3 | 0.3 | 0.5 | 0.6 | 0.4 | **0.2** |
| | | @ 0.2 | 0.5 | **0.4** | 6.0 | 2.8 | 0.7 | 2.4 | 8.1 | 0.8 | 1.9 |
| | Sub | @ 0.1 | 0.2 | **0.1** | 0.4 | 0.3 | 0.2 | 0.3 | 0.5 | 0.3 | 0.3 |
| | | @ 0.2 | **0.5** | **0.5** | 4.4 | 2.9 | 0.8 | 2.1 | 6.8 | 0.9 | 1.6 |
| | Info | @ 0.3 | 2.4 | 1.3 | **1.1** | 1.7 | 1.4 | 1.7 | 2.0 | 1.7 | 1.9 |
| | | @ 0.5 | 38.0 | 34.7 | **14.6** | 33.3 | 29.7 | 30.8 | 24.2 | 28.2 | 26.8 |
| **Inverse-transform** | Del | @ 0.1 | 0.3 | 0.9 | 0.6 | 0.2 | **0.1** | **0.1** | 1.0 | 0.3 | 1.6 |
| | | @ 0.2 | **1.6** | 10.3 | 6.9 | 2.7 | 1.7 | 1.9 | 9.0 | 3.6 | 14.5 |
| | Sub | @ 0.1 | 0.2 | 0.3 | **0.1** | **0.1** | **0.1** | **0.1** | 0.2 | **0.1** | 0.2 |
| | | @ 0.2 | 0.5 | 1.5 | 1.0 | **0.4** | **0.4** | **0.4** | 1.3 | 0.6 | 2.6 |
| | Info | @ 0.3 | 2.5 | 1.6 | **0.0** | 1.3 | 0.9 | 1.6 | 0.1 | 0.2 | 0.1 |
| | | @ 0.5 | 34.6 | 33.7 | **0.4** | 19.6 | 16.4 | 26.1 | 0.7 | 2.2 | 3.1 |
| **SynthID** | Del | @ 0.1 | 2.9 | **2.0** | 3.2 | 2.4 | **2.0** | 2.5 | 4.7 | 2.1 | 2.1 |
| | | @ 0.2 | **6.0** | 7.8 | 16.9 | 9.8 | 6.1 | 9.0 | 24.4 | 7.2 | 12.6 |
| | Sub | @ 0.1 | 2.3 | 1.6 | 2.7 | 1.8 | **1.5** | 1.9 | 4.1 | 1.6 | 1.8 |
| | | @ 0.2 | 5.5 | 5.9 | 15.3 | 8.7 | **5.1** | 7.9 | 21.9 | 6.0 | 11.5 |
| | Info | @ 0.3 | 16.6 | 14.8 | 0.1 | 2.6 | 1.9 | 4.2 | 0.5 | 0.6 | **0.0** |
| | | @ 0.5 | 94.5 | 83.8 | **0.1** | 5.4 | 9.5 | 23.6 | 0.6 | 0.6 | 0.9 |

## 4.3 Results on Robustness

Finally, we present the robustness evaluation of different GoF tests, focusing on their performance under various editing scenarios. To encourage diversity and reduce repetition, the temperature is fixed at 1. For all experiments, the critical values (or thresholds) are set at a significance level of $\alpha = 0.01$, and we evaluate how the Type II error varies across different edit types and intensities. We report the Type II error at a token length of 250 for word deletion and 300 for synonym substitution and information-rich edits. Table 3 presents the Type II errors for watermarked texts, averaged across three LLMs on the `C4` dataset. Results for `ELI5` dataset are provided in Table 11 in the Appendix. The notation "editing method @ a fraction" indicates that the editing method modifies the specified fraction of watermarked text or pivotal statistics. The observations are as follows.

**Robustness against normal text edits.** GoF tests show strong robustness against word deletion and synonym substitution edits. For all watermarks, GoF tests perform as well as or better than the best baseline across all edit types and intensities. Specifically, under 20% word deletion, for the Inverse-transform and SynthID watermarks, although the baseline achieves the lowest error, `And` trails by just 0.1%. Notably, among the considered GoF tests, there is no single method that consistently outperforms others across all scenarios. For instance, `Phi` has the strongest robustness in the Gumbel-max watermark, while `And` outperforms it in the Inverse-transform watermark.

**Robustness against information-rich edits.** GoF tests are designed to detect global differences between the empirical CDF and the null $\mu_0$, rather than relying on a few extreme values. As a result, removing a few of the largest pivotal statistics—while impactful for sum-based baselines—does not significantly alter the overall shape of the empirical CDF, allowing GoF tests to remain effective. This robustness is evident under information-rich edits, where GoF tests consistently outperform baselines. For instance, `Kui`, `Wat`, and `Chi` maintain low Type II errors across all watermarking schemes, even when 30% or 50% of the tokens are selectively modified. In contrast, sum-based methods often degrade under such edits because they depend heavily on high-magnitude statistics. A clear example is the Gumbel-max watermark: Aaronson's score $h_{ars} = -\log(1 - y)$ is particularly sensitive to values of $y$ near one. Since information-rich edits specifically target and remove these high values,

Table 4: Usage cases of GoF tests.

| Usage Cases | Notes | Tests | Practical Scenarios |
|---|---|---|---|
| Low temperature with repetition | Repetition is more common at low temperatures and shifts the empirical CDF, which GoF tests exploit effectively. | And, Chi. | Code generation (see Appendix D.2 for more discussion) |
| High temperature | Strong watermark signals make deviations from $\mu_0$ more detectable, improving the power of GoF tests. | All general GoF tests. | Open-ended text generation. |
| Common text edits | GoF tests remain effective despite moderate edits, but no single test consistently dominates. | Test choice varies. | Homework detection. |
| Information-rich edits | GoF tests are less sensitive to the removal of extreme values, preserving robustness. | Kui, Wat, Chi. | Internal API leakage. |

this method's effectiveness is sharply reduced. GoF tests, by focusing on overall distributional shifts, avoid this pitfall and thus offer more reliable performance in adversarial settings.

## 4.4 Evaluation Summary

We conclude this section with a brief summary of our key findings. First, *GoF tests are highly effective at detecting watermark signals at high temperatures*. With a high temperature (e.g., $0.7$), they consistently achieve very low Type II error rates regardless of the text lengths. Second, *text repetition increases the deviation between the empirical CDF and the null distribution $\mu_0$, enabling GoF tests to outperform baseline methods in low-temperature settings*. Unlike baseline methods, GoF tests can effectively leverage these distribution-level deviations. Third, *GoF tests are robust to common edits and information-rich edits*. Their strength lies in evaluating overall distributional fit, rather than relying on individual extreme pivotal statistics, making them more resilient to various types of edits. Table 4 summarizes the practical usage of GoF tests.

## 5 Conclusion and Discussion

In this work, we propose the use of goodness-of-fit (GoF) tests for watermark detection and systematically evaluate eight GoF methods across three popular watermarking schemes. Through extensive experiments across models, datasets, temperature settings, and editing types, we find that GoF tests can match or even outperform existing baselines in both detection power and robustness. We also identify text repetition as a key factor that enhances GoF performance at low temperatures.

Our findings open several new directions for improving watermark detection with statistical tools. First, a deeper theoretical understanding is needed to explain why certain GoF tests perform better in specific scenarios—potentially building on the statistical framework introduced by Li et al. [33]. Second, adaptive detection strategies that dynamically select among GoF tests based on input characteristics could further enhance performance in practice. Finally, while our work focuses on scalar pivotal statistics, some detection rules, such as Dathathri et al. [9], have explored high-dimensional statistics. Applying GoF tests in high-dimensional settings remains challenging due to the curse of dimensionality. Future work could investigate approaches such as maximum mean discrepancy or dimensionality reduction to extend GoF-based detection to these more high-dimensional scenarios.

## Acknowledgments

This work was supported in part by NIH grants R01AG071174, U01CA274576, R01EB036016, R01EB037101, U01AG066833, R01LM014731, and P30AG073105, NSF grant DMS-2310679, a Meta Faculty Research Award, and Wharton AI for Business. The content is solely the responsibility of the authors and does not necessarily represent the official views of the NIH.

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

# Appendices

## A   Introduction of Considered Watermarks

### A.1   Gumbel-max Watermark

The Gumbel-max watermark is the first unbiased watermark [1], which has been implemented internally at OpenAI [41] and serves as a baseline in many studies. This watermark builds on the Gumbel-max trick, a sampling technique for multinomial distributions [17, 35, 25]. The trick first generates a random vector $\zeta = (U_w)_{w \in \mathcal{W}}$ consisting of $|\mathcal{W}|$ i.i.d. copies of $U(0,1)$ and guarantees that $\arg\max_{w \in \mathcal{W}} P_w^{-1} \log U_w$ follows the distribution $\boldsymbol{P} \equiv (P_w)_{w \in \mathcal{W}}$. Recognizing this fact, Aaronson [1] proposed the following decoder:

$$\mathcal{S}^{\mathrm{gum}}(\boldsymbol{P}, \zeta) := \arg\max_{w \in \mathcal{W}} \frac{\log U_w}{P_w}, \tag{5}$$

which is, by definition, unbiased [13, 43, 61, 31].

The pivotal statistic is $Y_t = Y(w_t, \zeta_t) = U_{t,w_t}$, where $\zeta_t = (U_{t,w})_{w \in \mathcal{W}}$ collects all pseudorandom numbers and $U_{t,w_t}$ is the entry corresponding to the selected token $w_t$. Without a watermark, the $U_{t,w_t}$'s are i.i.d. from $U(0,1)$, so $Y_t \sim \mu_0 = U(0,1)$. In contrast, if a watermark is present, (5) makes tokens with larger pseudorandom numbers more likely to be selected. Indeed, we have $Y_t \mid \boldsymbol{P}_t \sim \mu_{1, \boldsymbol{P}_t}$, where $\mu_{1, \boldsymbol{P}}(Y \leq r) = \sum_{w \in \mathcal{W}} P_w r^{1/P_w}$ for $r \in [0,1]$ [31]. This alternative distribution differs from $\mu_0$ unless $\boldsymbol{P}$ is degenerate (i.e., concentrated on a single token).

A sum-based detection rule is defined by a scalar score function $h$. The watermark is claimed when the sum-based statistic $T_n = \sum_{t=1}^{n} h(Y_t)$ exceeds a chosen threshold. Existing scoring functions are listed in Table 5. Examples include Aaronson's score $h_{\mathrm{ars}}(y) = \log \frac{1}{1-y}$, the log function $h_{\log}(y) = \log y$ [29, 13], and the optimal least-favorable score $h_{\mathrm{lst}}(y) = \frac{\mathrm{d}\mu_{1, \boldsymbol{P}_\Delta^\star}(y)}{\mathrm{d}\mu_0(y)}$ from Li et al. [31], where $\Delta$ is a user-specified regularity parameter and $\boldsymbol{P}_\Delta^\star := (1 - \Delta, \Delta, 0, \ldots)$ is the least favorable NTP distribution. The function $h_{\mathrm{lst}}$ is minimax optimal when each $\boldsymbol{P}_t$ belongs to a $\Delta$-regular class, denoted $\mathcal{P}_\Delta$, defined as:

$$\mathcal{P}_\Delta = \{\boldsymbol{P} : \max_{w \in \mathcal{W}} P_w \leq 1 - \Delta\}. \tag{6}$$

Li et al. [31] shows that $h_{\mathrm{lst}}$ achieves the fastest exponential decay of Type II error at a fixed significance level $\alpha$ when testing against the least favorable distribution in $\mathcal{P}_\Delta$.

Table 5: Baseline sum-based detection rules. Since `Lst` can be applied to all watermarking schemes, we tune the parameter $\Delta$ for each to ensure good performance. In our experiments, we set $\Delta = 0.2$ for both the Gumbel-max and SynthID watermarks, and $\Delta = 0.5$ for the Inverse transform watermark.

| Tests name | Score function | Notation |
|---|---|---|
| Least-favorable test [31] | $h_{\mathrm{lst}}(y) = \frac{\mathrm{d}\mu_{1, \boldsymbol{P}_\Delta^\star}(y)}{\mathrm{d}\mu_0(y)}$ | `Lst` |
| Aaronson's score [1] | $h_{\mathrm{ars}}(y) = \log \frac{1}{1-y}$ | `Ars` |
| Logarithmic score [29] | $h_{\log}(y) = \log(y)$ | `Log` |
| Negative-sum score [29] | $h_{\mathrm{neg}}(y) = -y$ | `Neg` |
| Identity-sum score [9] | $h_{\mathrm{id}}(y) = y$ | `Sum` |

### A.2   Inverse Transform Watermark

The inverse transform watermark, proposed by Kuditipudi et al. [29], is another unbiased watermarking scheme. It uses a pseudorandom variable $\zeta = (U, \pi)$, where $U \sim U(0,1)$ is a uniform random variable and $\pi$ is an independent random permutation over the vocabulary $\mathcal{W}$. The decoder is defined via inverse transform sampling:

$$\mathcal{S}^{\mathrm{inv}}(\boldsymbol{P}, \zeta) := \pi^{-1} \circ F_\pi^{-1}(U),$$

where $F_\pi(\cdot)$ denotes the CDF under the permutation $\pi$, defined as $F_\pi(x) = \sum_{w' \in \mathcal{W}} P_{w'} \cdot \mathbf{1}_{\pi(w') \leq x}$.

By construction, the inverse transform watermark is unbiased for sampling from the NTP distribution $\boldsymbol{P}$. To see the unbiasedness directly, for any token $w$, note that $\mathcal{S}^{\text{inv}}(\boldsymbol{P}, \zeta) = w$ if and only if

$$\sum_{w' \in \mathcal{W}} P_{w'} \cdot \mathbf{1}_{\{\pi(w') < \pi(w)\}} \leq U \leq \sum_{w' \in \mathcal{W}} P_{w'} \cdot \mathbf{1}_{\{\pi(w') \leq \pi(w)\}}, \tag{7}$$

which occurs with probability $P_w$ since the interval above has length $P_w$.

The associated pivotal statistic is $Y(w, \zeta) = 1 - |U - \eta_\pi(w)|$, where $\eta_\pi(w) = \frac{\pi(w)-1}{|\mathcal{W}|-1}$ is the normalized rank of $w$ in the permuted vocabulary. Under $H_0$, $Y$ follows a distribution with CDF $r^2$ for $r \in [0, 1]$. Under $H_1$, the distribution is more complex, though closed-form approximations are available in certain asymptotic regimes [31]. Detection is performed using a sum-based rule with score function $h_{\text{neg}}(y) = -y$ [29].

### A.3 SynthID Watermark

The SynthID watermark, proposed by Google [9], is based on a novel sampling strategy for categorical distributions called tournament sampling. Given a number of tournament rounds $k$, the pseudorandom seed is defined as $\zeta = (\boldsymbol{g}_i)_{i=1}^k$, where each $\boldsymbol{g}_i = (g_{i,w})_{w \in \mathcal{W}}$ is a vector of i.i.d. samples drawn from $U(0, 1)$. Unlike the previous two watermarking schemes, the decoder here is stochastic rather than deterministic. The next token is sampled from a modified NTP distribution, i.e., $w \sim \mathcal{S}^{\text{syn}}(\boldsymbol{P}, \zeta)$, where

$$\mathcal{S}^{\text{syn}}(\boldsymbol{P}, \zeta) = \mathcal{T}_{\boldsymbol{g}_k} \circ \cdots \circ \mathcal{T}_{\boldsymbol{g}_1}(\boldsymbol{P}), \tag{8}$$

where $\mathcal{T}_{\boldsymbol{g}}$ is a vectorized operator defined as

$$(\mathcal{T}_{\boldsymbol{g}}(\boldsymbol{P}))(w) = P_w \cdot \left( P_w + 2 \cdot \sum_{w': g_{w'} < g_w} P_{w'} \right). \tag{9}$$

Dathathri et al. [9] prove that the output of $\mathcal{T}_{\boldsymbol{g}}(\boldsymbol{P})$ corresponds to the distribution of the winner token in a one-versus-one competition, where two independent tokens sampled from $\boldsymbol{P}$ compete based on their assigned $\boldsymbol{g}$ values. Repeating this process over $k$ rounds with i.i.d. pseudorandom vectors $\boldsymbol{g}_i$ results in $\boldsymbol{P}_\zeta$ being the distribution of the final tournament winner after $k$ rounds of competition.

For detection, we recover the $g$-values $\zeta = (\boldsymbol{g}_i)_{i=1}^k$ and extract the coordinates corresponding to the selected token $w$, i.e., $\{g_{i,w}\}_{i \in [k]}$. Under the tournament sampling scheme, these values tend to be larger under $H_1$ than under $H_0$, due to the winner-takes-all interpretation. We consider the sum-based detection rule proposed by Dathathri et al. [9] as our baseline, which computes a simple average over the selected $k$ $g$-values:

$$Y^{\text{sum}}(w, \zeta) = \frac{1}{k} \sum_{i=1}^k g_{i,w}.$$

For simplicity, we use the identity score function in our implementation. Under $H_0$, each $g_{i,w}$ is an independent $U(0, 1)$ random variable, so $Y^{\text{sum}}(w, \zeta)$ follows a scaled Irwin–Hall distribution: $Y^{\text{sum}}(w, \zeta) \sim \frac{1}{k}\text{IrwinHall}(k)$. In their paper, $k = 30$.

Note that Dathathri et al. [9] also proposed a weighted sum detection rule; however, we do not consider it in our evaluation, as its null distribution $\mu_0$ does not admit a closed-form expression and its empirical performance closely matches that of the unweighted sum statistic $Y^{\text{sum}}$, especially at high temperatures. In addition, they propose a Bayesian detector based on a neural network trained to distinguish human-written from watermarked text. We exclude this baseline as well, since its null distribution is analytically intractable due to the network's complex dependence on both the token $w$ and the randomness $\zeta$.

### A.4 Green-red List Watermark

The green-red list watermark, introduced by Kirchenbauer et al. [27], is a simple yet effective scheme. It randomly partitions the vocabulary $\mathcal{W}$ into a green (favored) set and a red (disfavored) set. Given a green-list fraction $\gamma \in (0, 1)$, it selects a random subset $\mathcal{G} \subset \mathcal{W}$ of size $|\mathcal{G}| = \gamma|\mathcal{W}|$ as the green set; the remaining tokens form the red set.

The decoder is stochastic: the next token $w$ is sampled from a shifted version of the original NTP distribution $\boldsymbol{P}$, that is, $w \sim \mathcal{S}^{\mathrm{grl}}(\boldsymbol{P}, \zeta)$ where

$$\mathcal{S}^{\mathrm{grl}}(\boldsymbol{P}, \zeta)(w) = \begin{cases} \frac{P_w \exp(\delta)}{Z} & \text{if } w \in \mathcal{G}, \\ \frac{P_w}{Z} & \text{otherwise}, \end{cases}$$

and $Z$ is the normalization constant. The pseudorandom component $\zeta$ here is the green set $\mathcal{G}$.

For a given token, detection is based on whether it belongs to the green set: the pivotal statistic is $Y^{\mathrm{grl}}(w, \zeta) = \mathbf{1}_{w \in \mathcal{G}}$. Under $H_0$, $\mathcal{G}$ is independent of the token, so $Y^{\mathrm{grl}} \sim \mathrm{Ber}(\gamma)$. Under $H_1$, the decoder increases the likelihood of selecting green tokens, resulting in $Y^{\mathrm{grl}} \sim \mathrm{Ber}(\mu)$, where

$$\mu = \frac{\sum_{w \in \mathcal{G}} P_w \exp(\delta)}{\sum_{w \in \mathcal{G}} P_w \exp(\delta) + \sum_{w \notin \mathcal{G}} P_w}.$$

Since $\mu > \gamma$ in general, detection reduces to checking whether the number of green tokens $\sum_{t=1}^{n} Y_t^{\mathrm{grl}}$ exceeds a threshold.

Variants of this scheme, such as DiPmark [56], use more sophisticated partitions, but the core idea remains the same: secretly boost the probabilities of selected tokens and count how often they appear.

**Why we don't consider green–red list watermarks.** We did not include the green–red list watermark and its variants in experiments because GoF tests reduce to the original detection method in this setting. This reduction happens because the pivotal statistic (i.e., whether token is green) is binary (0 or 1), and the order of these values does not affect the test. As a result, the full information is captured by simply counting the number of 1s. In this case, testing whether the number of 1s exceeds a threshold (i.e., the original detection rule) is already highly effective, and both GoF tests and the original method rely on this same statistic—making them essentially equivalent. Therefore, additional experiments would be redundant.

# B Related Work

**Most related work.** Our work evaluates general GoF tests for watermark detection. The closest prior work is Li et al. [33], which proposed a parameterized GoF test specifically for the Gumbel-max watermark and established its statistical optimality under certain settings. In contrast, we study a broader class of non-parameterized GoF tests applicable to a wide range of watermarking schemes, offering greater flexibility. We also uncover new insights not included in [33]: the role of text repetition in watermark detection. While several prior studies [13, 28] have examined text repetition, they mainly focused on mitigation strategies or preprocessing methods. Our work is distinct in that we analyze repetition from a statistical perspective, focusing on how it affects the behavior of pivotal values and, in turn, the reliability of GoF-based detection compared to sum-based approaches. Through ablation studies, we show that GoF tests can exploit repetition—particularly in low-temperature decoding—to boost detection performance. This phenomenon, is not captured by Li et al. [33], offers a new understanding of how GoF tests succeed in practice.

**Other optimal detection rules.** Several works focus on optimal detection for specific watermarks. Huang et al. [22] derive optimal rules which, however, require exponential computation or access to full NTP distributions, limiting their practicality. Li et al. [31] assume all NTP distributions fall into a prior class of $\boldsymbol{P}_t$, and develop worst-case optimal detection rules using a minimax formulation. Huang et al. [23] study speculative sampling, which is beyond our scope. He et al. [18] propose a variant of the Gumbel-max watermark together with an optimal detection rule optimized for detection power, sample distortion, and Type-I error. While these methods target specific schemes, our focus is on general GoF tests, which can be applied broadly as long as the null distribution $\mu_0$ is tractable.

**General on watermarking schemes.** A wide range of watermarking schemes have been proposed in recent work [28, 13, 29, 21, 56, 61, 60, 34, 16, 14]. At a high level, these methods define pseudorandomized sampling procedures over next-token prediction (NTP) distributions, where the seeded randomness allows watermarked tokens to be reliably identified at test time. A decoder is considered unbiased if the watermarked token distribution exactly matches the original NTP

distribution; otherwise, it is biased [31]. In practice, each new watermarking scheme is often accompanied by a custom detection rule designed in an ad hoc manner. Many such examples are provided in Appendix A. In contrast, our work focuses on evaluating NTP-agnostic GoF tests as general detection tools. Our goal is to systematically assess their performance across diverse watermarking methods and to understand the key factors that influence their detection effectiveness.

**GoF tests in statistics and NLP.** GoF tests are standard tools for assessing whether i.i.d. samples follow a specified distribution [7]; see Section 3 for representative examples. These tests are particularly effective in low-dimensional settings and often rely on empirical CDFs or binned statistics. To our knowledge, we are the first to apply GoF tests to watermark detection. A key distinction is that watermark detection involves non-i.i.d. samples under $H_1$ due to the autoregressive nature of LLMs: each $P_t$ depends on previous tokens. Our work builds on the frameworks of Li et al. [31, 33] but shifts the focus toward broader applicability and robustness. GoF tests have also been used in NLP for analyzing language statistics [37] and evaluating significance in empirical studies [10]. Our work connects these statistical tools to the challenge of modern watermark detection.

## C  Practical Considerations for GoF Tests

**Asymptotic Properties of GoF Tests.** Most of the classical GoF tests considered in our work—Tr-GoF test (`Phi`), Kuiper (`Kui`), Kolmogorov–Smirnov (`Kol`), Anderson–Darling (`And`), Cramér–von Mises (`Cra`), and Watson (`Wat`)—are known to have only limiting distributions under the null hypothesis. That is, their exact finite-sample distributions are analytically intractable. In contrast, one of the baseline methods, Aaronson's score (`Ars`), admits an explicit Gamma distribution under the null, and prior work [13] has demonstrated that it can provide strong non-asymptotic guarantees for Type I error control. However, despite the asymptotic nature, we argue that GoF tests remain practically reliable for watermark detection. Our rationale is threefold:

- **Empirical reliability of Type I error control.** It's worth noting that the limiting distributions offer valuable theoretical tools, but in practice, simulations offer users a more flexible and convenient way to compute critical values—like in Python package SciPy[4], due to the complex forms of some distributions (e.g., the Anderson-Darling distribution [36, 2]). Even though critical values are obtained from simulations, our experiments (see Figure 4) show that these GoF tests consistently maintain Type I error rates (i.e., FPR) close to the nominal level, even at stringent settings (e.g., $\alpha = 10^{-5}$).

- **Validity of simulation-based calibration.** Simulation provides a principled and widely adopted approach for obtaining critical values when exact distributions are unavailable. By generating i.i.d. pivotal statistics under the null distribution $\mu_0$, one can estimate the empirical $\alpha-$quantile with arbitrarily small error by increasing simulation size. This calibration can be performed prior to deployment and reused across tasks.

- **Practical robustness in Type II error.** Beyond Type I control, we find that GoF tests retain strong detection power (i.e., low Type II error) under low $\alpha$ levels. Table 6 demonstrates that even when the Type I error is tightly constrained, GoF texts continue to outperform the baselines.

**Complementary Role of GoF Tests.** Importantly, we do not view GoF tests as a replacement for sum-based or non-asymptotic methods, but rather as complementary tools. While methods like `Ars` target high-value pivotal statistics, GoF tests are designed to detect distributional shifts, offering enhanced robustness especially under information-rich edits (see Section 4.3). Despite relying on asymptotic or simulated critical values, GoF tests enrich the detection toolkit by capturing distribution-level anomalies.

Our experiments show that GoF tests provide reliable Type I error control and strong detection power, yet their reliance on asymptotic or simulation-based thresholds means they may fall short in high-stakes settings–for example, legal proceedings where certifying false positive rates with exact finite-sample guarantees is required. In such scenarios, GoF tests alone are insufficient. More broadly, no single method should serve as the sole line of defense. A robust system should draw from multiple

---

[4]`https://docs.scipy.org/doc/scipy/reference/generated/scipy.stats.goodness_of_fit.html`

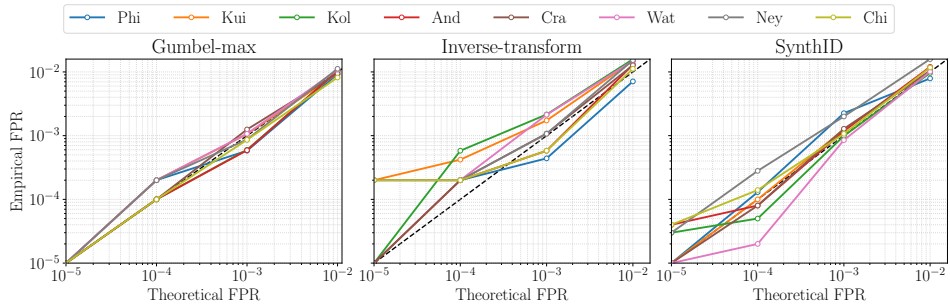

Figure 4: Theoretical vs. empirical false positive rates (FPR) for GoF tests. Results are based on 100k sampled human-written texts from the `C4` dataset, with repeated pivotals removed. Even at this scale, GoF tests maintain strong Type I error control, achieving empirical FPRs as low as the target significance level.

Table 6: Type II errors at a stricter Type I error level of $10^{-5}$. Results are reported for temperature $T = 0.7$ and sequence length $n = 400$, averaged over three LLMs on the `C4` dataset. `Baseline` denotes the best-performing method among all baseline detectors. The results show that GoF tests retain strong detection power even under tight Type I error constraints.

| Watermarks | Baseline | Phi | Kui | Kol | And | Cra | Wat | Ney | Chi |
|---|---|---|---|---|---|---|---|---|---|
| **Gumbel.** | 1.1 | 1.1 | 0.7 | 0.9 | 0.7 | 0.9 | 1.2 | 0.6 | **0.4** |
| **Inverse.** | 2.6 | 17.3 | 1.4 | **1.2** | **1.2** | 1.6 | 2.9 | 1.8 | 1.5 |
| **SynthID** | 4.2 | 18.0 | 4.6 | 4.1 | 3.9 | 4.7 | 11.8 | 4.1 | **2.7** |

methods, each with different strengths–for instance, `Ars` for sensitivity to large deviations and GoF tests for distributional anomalies.

# D    Experiment Details and Results

## D.1    Prompts for text generation

For the text completion task, we truncated the first 50 tokens directly, without adding any instructions, and used them as the prompt. For the long-form question-answering task, we manually prepended the instruction "*Answer the following question:*" before the question and used this as the prompt. To ensure uniform prompt lengths, we selected questions whose total length, including the instruction, was within 50 tokens. Additionally, we added padding tokens at the beginning to ensure the final prompt length reached 50 tokens. Table 7 shows example prompts for these two tasks.

Table 7: Example prompts for the text generation tasks.

| Text completion | Long-form question-answering |
|---|---|
| Prompt: The Arlington County Board plans to vote Saturday afternoon on giving Amazon 23 million and other incentives to build a headquarters campus in Crystal City, but only after hearing scores of northern Virginia residents and advocates testify for or against the project.\n The five-member | Prompt: [pad][pad]...[pad] Answer the following question: Why is it that we calm down when we take a deep breath, hold it for a few seconds and exhale? |

## D.2    Example of repetition

In this part, we provide two examples of repetitions in both human-written and watermarked text, where we use colored backgrounds to highlight the most repeated content. The first example comes from the `C4` dataset, which is an NBA game report in which the phrase "*during the second half*"

appears 15 times. As discussed in Section 4.2, such repetitions can introduce distinct jumps in the empirical CDF of pivotal statistics, potentially leading GoF tests to incorrectly reject the null hypothesis. The second example, generated by Llama 3.1-8B with a temperature of 0.7 using the Gumbel-max watermark for the text completion task, exhibits a more direct and deliberate pattern of repetition compared to human-written text.

*Example of repetition in human-written text:* Lakers guard Kobe Bryant flys to the basket with Rockets forward Shane Battier trailing during the second half. Rockets forward Luis Scola drives around the Lakers' Luke Walton during the second half. The Rockets' Aaron Brooks jumps up as the crowd goes wild after one of his three-point shots during the second half. The Lakers' Ron Artest battles with the Rockets' Chuck Hayes for a loose ball during the second half. Rockets guard Aaron Brooks drives around the Lakers' Derek Fisher and Lamar Odom during the second half. Rockets forward Trevor Ariza drives up the court against the Lakers' Lamar Odom during the second half. The Rockets' Kyle Lowry drives up the court against the Lakers' Derek Fisher during the second half. The Lakers' Ron Artest with his haircut during the second half. Lakers guard Kobe Bryant flys to the basket with Rockets forward Louis Scola trailing during the second half. Ropckets forward Trevor Ariza goes up for a basket during the second half. The Lakers' Kobe Bryant gets a shot off as the Rockets' Shane Battier tries to defend during the second half. The Lakers' Kobe Bryant bumps into the Rockets' Shane Battier during the second half. The Lakers' Kobe Bryant booed by fans during a free throw during the second half. Lakers forward Ron Artest fights Rockets forward Chuck Hayes for the ball in the fourth quarter. Lakers forward Ron Artest puts a hand in the face of Rockets forward Carl Landry during the second half. Rockets guard Trevor Ariza celebrates with Pops Mensah-Bonsu after his three-point shot during the last seconds of the second half. The Rockets' Trevor Ariza tries to get his hands on a ball held by the Lakers' Ron Artest during the second half.

*Example of repetition in LLM-written text:* Two slaughtermen have been sacked after an undercover investigation exposed shocking cruelty to horses at an abattoir. The disturbing video shows them being beaten with metal poles and illegally stunned in groups of up to three at a time before being killed. In one clip, one of the workers can be heard saying: 'We're going to kill them now. Don't beat them up. Don't beat them up.Don't beat them up.Don't beat them up.Don't beat them up.Don't beat them up...

**Repetition in more powerful LLMs.** Importantly, extensive repetition will become increasingly rare in newer and more powerful LLMs, even at low temperatures. For instance, closed-source models such as GPT and Claude are capable of producing diverse outputs under low-entropy (low-temperature) settings without excessive repetition. Thus, as summarized in Table 4, the specific low-temperature scenario with heavy text repetition should be regarded as an increasingly rare phenomenon in more advanced models. Accordingly, our analysis of the text repetition can be viewed as an edge-case study that *reveals how repetition influences the behavior of pivotal statistics and, in turn, the reliability of GoF tests compared to sum-based methods*. More generally, detection in low-entropy settings without text repetition remains a difficult problem for GoF tests, as it does for other methods, making it a challenging yet worthwhile direction for future research.

### D.3 Cost of GoF tests

The primary computational cost of goodness-of-fit (GoF) tests in watermark detection lies in calculating the critical values. Most GoF tests—including all those used in this work—lack non-asymptotic distributions but do have asymptotic ones. For tests with simpler asymptotic forms (e.g., `Ney` and `Chi`), the critical values can be computed directly from asymptotic results, leading to runtime comparable to sum-based baselines.

In contrast, several GoF tests (e.g., `Kui`, `Kol`, `And`, `Cra`, `Wat`) rely on Monte Carlo simulation for critical value computation due to the complexity of their asymptotic distributions. This substantially increases computational cost.

Importantly, once the watermark scheme is fixed, the null distribution is known prior to testing, enabling the critical values to be pre-computed and reused across runs. As a result, **the actual cost of applying GoF tests in practice is very low**, even for those tests that require Monte Carlo simulation.

Table 8 reports runtime for detecting 100 pivotal sequences with $n = 400$ and simulation size $10^5$. For comparison, the `Ars` baseline takes $1.30 \times 10^{-3}$ seconds. As shown, Monte Carlo–based GoF tests require roughly $10^3$ times longer than the baseline, primarily because of the $10^3$-fold larger simulation size needed for estimating critical values.

Table 8: Runtime cost of GoF tests for detecting 100 pivotal sequences with $n = 400$ and simulation size $10^5$. Time* indicates the time cost without calculating the critical value.

| Test | Time | Time* |
|------|------|-------|
| Phi | $2.11 \times 10^0$ | $2.20 \times 10^{-3}$ |
| Kui | $2.03 \times 10^0$ | $1.50 \times 10^{-3}$ |
| Kol | $1.95 \times 10^0$ | $1.40 \times 10^{-3}$ |
| And | $2.61 \times 10^0$ | $4.30 \times 10^{-3}$ |
| Cra | $2.37 \times 10^0$ | $1.70 \times 10^{-3}$ |
| Wat | $2.38 \times 10^0$ | $1.80 \times 10^{-3}$ |
| Ney | $4.00 \times 10^{-3}$ | same as left |
| Chi | $3.90 \times 10^{-3}$ | same as left |

### D.4 More experiment results

All text generation tasks were conducted on NVIDIA A100 GPUs, with a total computational cost of approximately 360 GPU hours to reproduce all experiments. Here, we present the results of additional experiments that were not included in the main content. Table 9 provides a list of additional results along with brief descriptions.

Table 9: List of additional results.

| Label | Brief description |
|-------|-------------------|
| Figure 5, 6, 7 | Type II errors for detection tests applied to the Gumbel-max, Inverse-transform and SynthID watermark. |
| Figure 8, 9 | Percentage of repetition in watermarked texts compared to human-written texts. Distribution of the highest probability in NTP distributions. |
| Table 10 | Type II errors on the ELI5 dataset. |
| Table 11 | Type II errors on the ELI5 dataset under various types of edits. |

## E   Additional Discussions

### E.1   The influence of different LLMs

Different LLMs may produce different next-token prediction distributions under the same prompt, which can impact the generation of watermarked text. In our experiments, we apply three popular open-source LLMs: OPT-1.3B, OPT-13B [59], and Llama 3.1-8B [11]. These models vary in their text generation capabilities, as indicated by their rankings on Hugging Face's Open LLM Leaderboard [5], with OPT-1.3B < OPT-13B < Llama 3.1-8B in terms of overall performance. In our settings, OPT models illustrate scaling effects within a family, while Llama 3.1-8B provides a contrast with a newer architecture and training paradigm, giving us both comparability and diversity in model selection.

Across all detection tests, we observe some variation in performance among the three LLMs, but the differences are not substantial. More importantly, the relative performance trends between tests remain similar across models (see Figure 5, 6, 7). While specific rankings may fluctuate slightly, tests that perform well on one LLM generally show comparable strength on the others. For example, Chi consistently achieve strong performance at low temperatures across all three LLMs. In other words, while absolute performance varies by model, the overall patterns in test effectiveness are largely preserved across different LLMs.

While our analysis is limited to three representative open-source models, the consistency of results across both within-family scaling (OPT-1.3B vs. OPT-13B) and cross-family comparison (OPT vs.

---

[5] https://huggingface.co/open-llm-leaderboard

Llama 3.1) provides encouraging evidence that the relative effectiveness of GoF tests is not highly sensitive to model choice. Architectural and training differences may lead to further variation in other settings, but our findings suggest that the key performance patterns we identify are likely to extend beyond the specific models studied here. Moreover, even if the performance of a specific test does not transfer directly to a new setting, the low computational cost of these GoF tests (see Appendix D.3) allows practitioners to easily run all suggested tests and select the best-performing one in just a few minutes. This makes our approach flexible and broadly applicable, while further evaluations across additional model families and domains can continue to strengthen these conclusions.

### E.2 The influence of different text generation tasks

LLM performance varies by task, potentially impacting text generation quality. For example, Question-Answering requires a stronger ability for LLM to generate valid responses than Text Completion [3]. From our experiments, we found that task type has a greater effect on detection test performance than differences between LLMs. Specifically, as shown in Figures 5, 6 and 7, baseline methods exhibit noticeable performance drops on the ELI5 dataset at temperatures 0.1, 0.3, and 0.7, while GoF tests demonstrate stronger effectiveness on it. This can be attributed to the influence of repetition, as discussed in Section 4.2. LLMs tend to generate more repetitions in long-form question-answering tasks compared to text completion, especially at low temperatures. This benefits GoF tests while diminishing the effectiveness of sum-based baselines. Additionally, we observe a high occurrence of ineffective outputs in long-form question-answering at low temperatures. To take an extreme example, when answering the question: "*Why does 1080p on a 4K TV look better than 1080p on a 1080p TV?*", LLaMA 3.1-8B generates: "*I; 1;1;1;1;1;1;1;1;...*" with temperature set to 0.1. Two factors may contribute to this issue. First, the LLMs we applied may not be strong enough to generate valid answers for certain questions. Second, some questions inherently lack sufficient content for an extended response, even for human writers.

## F  Broader Impacts

This paper investigates the use of classic goodness-of-fit (GoF) tests for detecting text watermarks in LLM-generated outputs, aiming to enhance the reliability and robustness of watermark detection. A potential positive societal impact is the promotion of content authenticity and provenance in an era of rapidly growing AI-generated media. Stronger watermark detection techniques can support efforts in combating misinformation, safeguarding intellectual property, and promoting transparency in automated content creation.

However, there are also potential negative societal impacts. Improved detection capabilities might encourage more widespread use of watermarks, including those that could be embedded without user consent, raising privacy and surveillance concerns. Additionally, the detection tools could be misused to reveal the use of LLMs in sensitive or anonymized content, possibly compromising user confidentiality.

To mitigate these risks, it is important that watermark detection methods are deployed transparently and ethically, with clear communication about their use and limitations. In conclusion, while our work on improving watermark detection for language models has the potential to strengthen content attribution and support responsible AI use, it is important to carefully consider and address potential negative societal impacts to ensure the technology is deployed ethically and transparently.

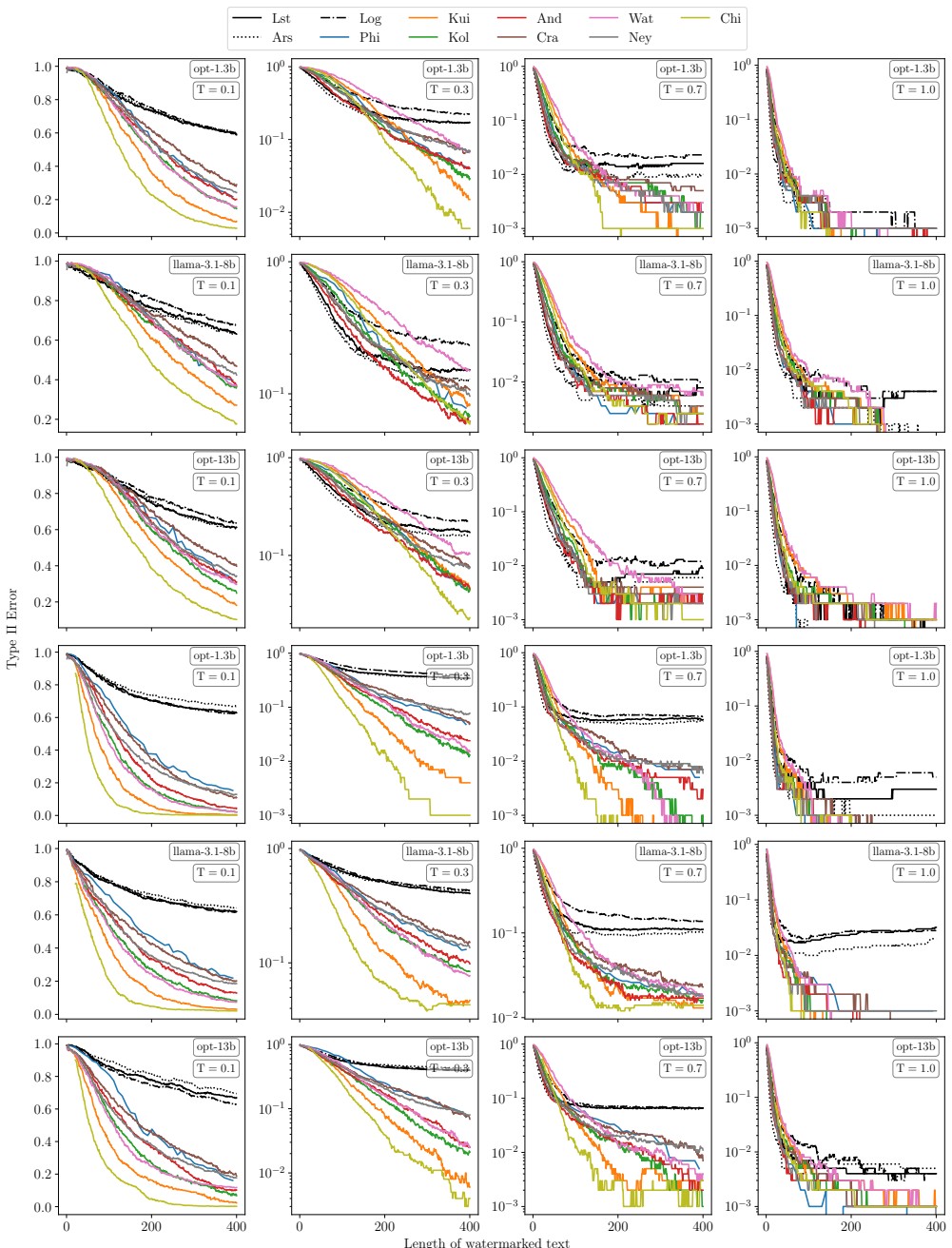

Figure 5: Empirical Type II errors for various detection tests applied to the Gumbel-max watermark across the C4 dataset (top three rows) and the ELI5 dataset (bottom three rows).

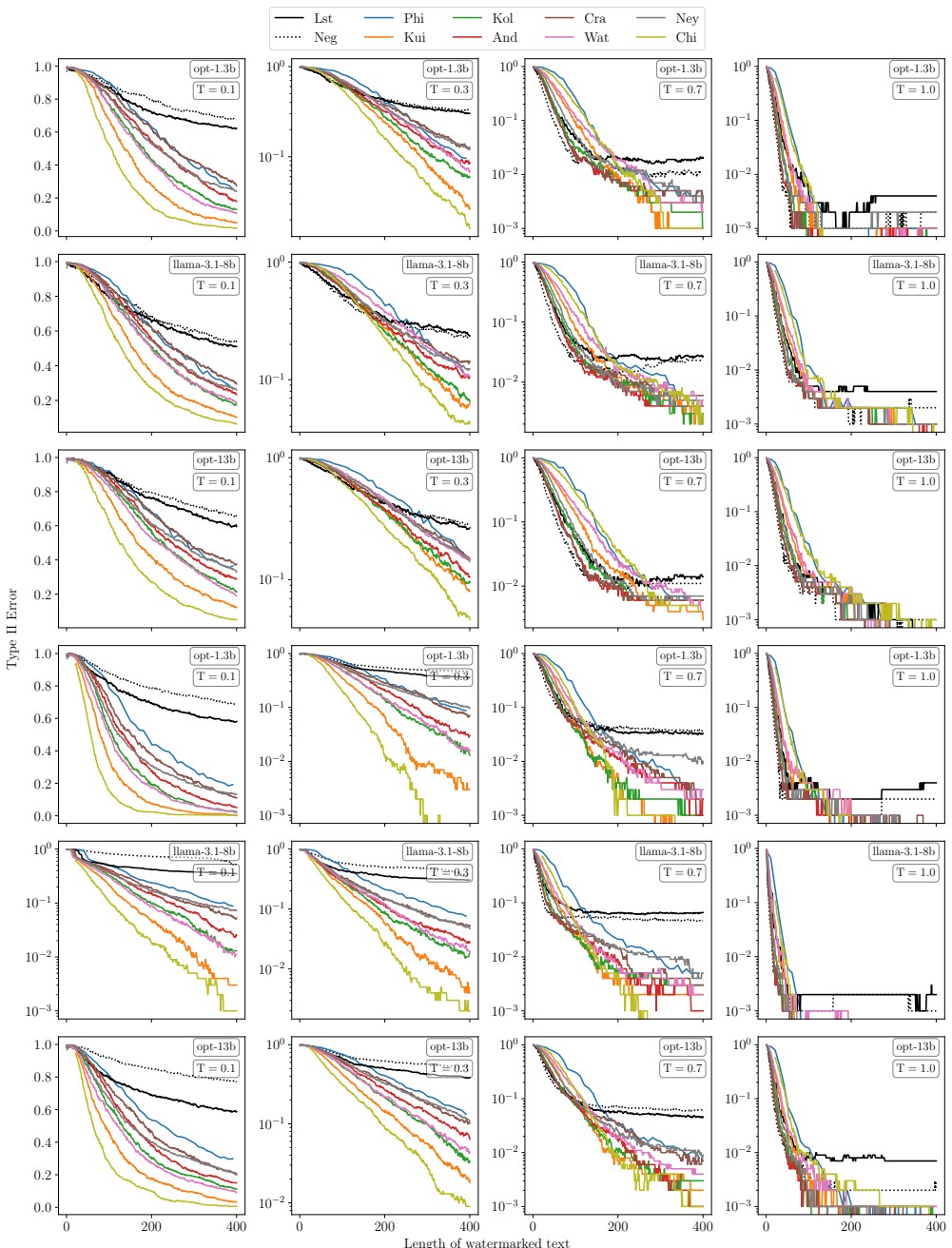

Figure 6: Empirical Type II errors for various detection tests applied to the Inverse-transform watermark across the `C4` dataset (top three rows) and the `ELI5` dataset (bottom three rows).

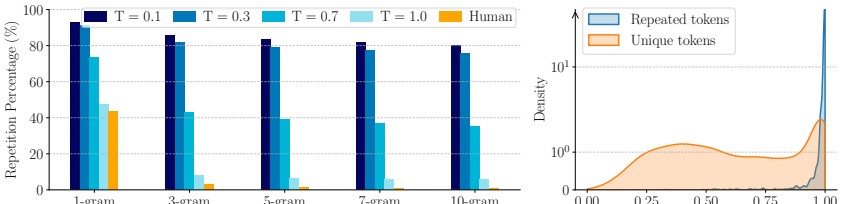

Figure 7: Empirical Type II errors for various detection tests applied to the SynthID watermark across the `C4` dataset (top three rows) and the `ELI5` dataset (bottom three rows).

Figure 8: **Left:** Average repetition rate in Inverse-transform watermarked texts across three LLMs, compared to human-written text. **Right:** Distribution of the highest probability in NTP distributions for OPT-1.3B using the Inverse-transform watermark at temperature 0.7.

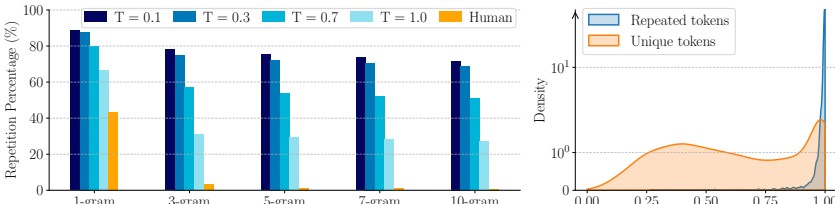

Figure 9: **Left:** Average repetition rate in SynthID watermarked texts across three LLMs, compared to human-written text. **Right:** Distribution of the highest probability in NTP distributions for OPT-1.3B using the SynthID watermark at temperature 0.7.

Table 10: Type II errors (averaged over three LLMs) on the `ELI5` dataset. All values are enlarged by 100 for readability. T denotes temperature and $n$ the token length. `Baseline` refers to the best-performing method among the baseline detectors. Red shading highlights lower values; blue indicates higher values.

|  | T | $n$ | Baseline | Phi | Kui | Kol | And | Cra | Wat | Ney | Chi |
|---|---|---|---|---|---|---|---|---|---|---|---|
| **Gumbel.** | 0.3 | 200 | 46.2 | 23.8 | 8.1 | 15.3 | 18.5 | 25.2 | 16.8 | 21.6 | **3.5** |
|  |  | 400 | 39.9 | 8.4 | 1.9 | 3.9 | 4.9 | 9.0 | 3.9 | 9.8 | **1.6** |
|  | 0.7 | 200 | 7.0 | 2.6 | 1.0 | 1.6 | 1.7 | 2.7 | 2.2 | 2.3 | **0.5** |
|  |  | 400 | 7.4 | 0.9 | **0.5** | 0.6 | 0.7 | 1.2 | 0.7 | 1.2 | 0.6 |
| **Inverse.** | 0.3 | 200 | 44.4 | 33.0 | 8.8 | 15.0 | 19.8 | 26.1 | 16.1 | 24.3 | **3.9** |
|  |  | 400 | 35.0 | 9.9 | 0.8 | 2.2 | 4.1 | 7.5 | 2.6 | 8.4 | **0.4** |
|  | 0.7 | 200 | 5.3 | 2.8 | 0.6 | 0.7 | 1.2 | 2.2 | 1.3 | 2.1 | **0.3** |
|  |  | 400 | 4.8 | 0.5 | **0.1** | 0.2 | **0.1** | 0.4 | 0.3 | 0.8 | **0.1** |
| **SynthID** | 0.3 | 200 | 60.4 | 40.6 | 21.5 | 30.7 | 34.0 | 41.8 | 31.0 | 39.2 | **11.5** |
|  |  | 400 | 49.3 | 15.6 | 5.8 | 9.9 | 13.4 | 19.0 | 10.2 | 18.8 | **3.2** |
|  | 0.7 | 200 | 6.1 | 4.5 | 4.1 | 3.1 | **2.9** | 3.9 | 7.5 | 4.2 | **2.9** |
|  |  | 400 | 5.5 | 1.5 | 1.2 | 1.0 | **0.8** | 1.2 | 2.3 | 1.5 | 1.2 |

Table 11: Type II errors (averaged over three LLMs) on the `ELI5` dataset with temperature 1.0 under various editing types. "Del" denotes word deletion, "Sub" represents synonym substitution, and "Info" refers to information-rich edits. All values are enlarged by 100 for readability. `Baseline` refers to the best-performing method among the baseline detectors. Red shading highlights lower values; blue indicates higher values.

|  | Edits |  | Baseline | Phi | Kui | Kol | And | Cra | Wat | Ney | Chi |
|---|---|---|---|---|---|---|---|---|---|---|---|
| **Gumbel-max** | Del | @ 0.1 | 1.1 | 0.3 | 0.5 | 0.6 | 0.4 | 0.8 | 1.2 | 0.5 | **0.2** |
|  |  | @ 0.2 | 1.6 | **0.8** | 4.4 | 2.7 | 1.3 | 2.8 | 6.3 | 1.2 | 1.5 |
|  | Sub | @ 0.1 | 1.3 | 0.3 | **0.2** | 0.4 | 0.3 | 0.7 | 0.5 | 0.3 | **0.2** |
|  |  | @ 0.2 | 1.5 | 0.6 | 3.2 | 2.4 | 1.0 | 2.4 | 4.7 | 1.1 | 0.9 |
|  | Info | @ 0.3 | 5.2 | 5.0 | **1.4** | 2.0 | 2.0 | 2.4 | 1.9 | 1.9 | **1.4** |
|  |  | @ 0.5 | 29.6 | 44.2 | **12.1** | 22.0 | 21.6 | 22.7 | 18.6 | 21.3 | 17.5 |
| **Inverse-transform** | Del | @ 0.1 | 0.8 | 0.9 | 0.5 | **0.3** | 0.4 | 0.5 | 1.0 | 0.6 | 1.1 |
|  |  | @ 0.2 | 2.5 | 6.1 | 5.3 | 2.6 | **2.0** | 2.5 | 7.2 | 3.2 | 10.2 |
|  | Sub | @ 0.1 | 0.7 | 0.6 | **0.2** | **0.2** | **0.2** | 0.3 | 0.3 | 0.3 | 0.3 |
|  |  | @ 0.2 | 1.0 | 1.4 | 1.0 | **0.4** | 0.5 | 0.5 | 1.1 | 0.6 | 1.5 |
|  | Info | @ 0.3 | 2.3 | 6.2 | **0.1** | 1.6 | 0.9 | 1.7 | 0.2 | 0.3 | **0.1** |
|  |  | @ 0.5 | 20.7 | 50.9 | **0.3** | 10.5 | 11.5 | 14.5 | 0.9 | 1.7 | 2.2 |
| **SynthID** | Del | @ 0.1 | 2.3 | 1.2 | 1.0 | 1.0 | 1.1 | 1.7 | 2.2 | 1.4 | **0.6** |
|  |  | @ 0.2 | 4.8 | 4.6 | 11.9 | 6.8 | **4.1** | 6.2 | 17.9 | 4.8 | 7.0 |
|  | Sub | @ 0.1 | 2.5 | 1.1 | 0.8 | 1.0 | 1.3 | 2.0 | 1.8 | 1.4 | **0.5** |
|  |  | @ 0.2 | 5.6 | 4.8 | 9.6 | 6.5 | **4.1** | 6.4 | 14.1 | 4.7 | 6.5 |
|  | Info | @ 0.3 | 8.9 | 4.9 | **0.0** | 0.7 | 0.5 | 1.7 | 0.1 | 0.6 | **0.0** |
|  |  | @ 0.5 | 81.3 | 64.3 | **0.1** | 6.9 | 7.0 | 14.9 | 0.4 | 0.7 | 0.5 |

