# OpenReview forum: "On the Empirical Power of Goodness-of-Fit Tests in Watermark Detection"
_NeurIPS.cc/2025/Conference — NeurIPS 2025 spotlight_

### Official Review · Reviewer_jT6m · 2025-06-23

**Clarity:** 4
**Significance:** 3
**Originality:** 2
**Rating:** 5
**Confidence:** 4

**Summary:**

The authors explore the effectiveness of goodness-of-fit (GoF) tests for detecting watermarks in text generated by LLMs. The authors evaluate eight distinct GoF tests across three popular watermarking schemes, utilizing various open-source LLMs, datasets, and text generation conditions. They find that GoF tests generally outperform existing detection methods in both accuracy and resilience, particularly in scenarios involving text repetition at lower temperatures or strong watermark signals at higher temperatures. The study highlights that GoF tests, by analyzing the full empirical distribution of pivotal statistics rather than just sum-based measures, offer a robust and underused tool for verifying the origin of LLM-generated content, even when modified.

**Questions:**

- How expensive are the GoF tests compared to the existing detection methods? Appendix C3 doesn't provide enough details for readers to have an idea of the tradeoff between effectiveness and efficiency.
- Check this [paper](https://arxiv.org/abs/2405.20777) out. They also make use of statistical tests to detect watermarks. Try to compare their proposed methods especially the robustness to the same attacks you evaluated.

**Ethical Concerns:**

["NO or VERY MINOR ethics concerns only"]

**Final Justification:**

After considering the rebuttal and discussion, I am increasing my score due to the paper’s strong empirical contribution and its potential impact on future work in watermark detection.

**Resolved issues:**

* The authors clarified why no single GoF test is optimal and reframed their work as a toolkit with summarized use cases, which addresses my concern about practical applicability.
* The explanation of low-quality outputs at low temperatures (rare and minimally impactful) and the statistical stability of results (large sample Bernoulli trials) resolves concerns about robustness and significance reporting.
* Additional details on computational trade-offs between tests and clarification of differences from related black-box work (e.g., Gloaguen et al.) addressed scope-related ambiguities.

**Remaining issues:**

* Adaptive selection strategies and detailed runtime comparisons remain future work, limiting immediate deployment guidance.
* Some concerns about pathological low-temperature outputs could be better addressed with filtering or explicit sensitivity analysis, though this is minor.

These unresolved points are secondary; the main contribution is significant and stands out in the current literature. The work is methodologically solid and opens clear avenues for future research.

Given these factors, I now lean toward an accept.

**Limitations:**

Yes

**Quality:**

4

**Strengths And Weaknesses:**

**Strengths**
- The paper conducts a highly thorough evaluation, systematically testing eight GoF methods across three popular watermarking schemes (Gumbel-max, Inverse Transform, SynthID), using three open-source LLMs (OPT-1.3B, OPT-13B, Llama 3.1-8B), two datasets (C4, ELI5), four temperature settings, and multiple post-editing methods (deletion, substitution, information-rich edits). This extensive setup provides strong empirical backing for the claims.
- GoF tests are shown to consistently outperform traditional sum-based baseline methods in both detection power and robustness across varying temperatures and text lengths. The study also demonstrates that GoF tests maintain high detection power even under various text modifications, including common edits (word deletion, synonym substitution) and more adversarial information-rich edits. This is attributed to their focus on overall distributional shifts rather than reliance on extreme values.
- The paper is logically organized with a clear problem introduction, detailed preliminaries on watermarking and pivotal statistics, a clear explanation of GoF tests, comprehensive experimental sections, and a concise conclusion. The hypothesis testing framework for watermark detection and the role of pivotal statistics are clearly defined and explained, making the core problem accessible.
- The study is significant and novel. The paper is the first to systematically apply and evaluate a broad class of classical GoF tests for watermark detection, differentiating itself from prior work that focused on specific parameterized GoF tests. Also, the identification and analysis of text repetition as a factor that uniquely benefits GoF test performance in low-temperature settings is a significant original contribution. This provides a new understanding of detection effectiveness in challenging scenarios.

**Weaknesses**
- While GoF tests generally outperform baselines, the paper notes that "there is no single method that consistently outperforms others across all scenarios" within the GoF family . This suggests that practical deployment might require adaptive selection strategies, which is acknowledged as future work but remains a current limitation for straightforward application.
- In some low-temperature, long-form question-answering scenarios, the LLMs produced highly repetitive or ineffective low-quality outputs (e.g., "I; 1;1;1;1;1;1;1;1;1;...") . While the paper discusses potential reasons (LLM capability, question content), these instances might affect the representativeness or generalizability of results in such extreme low-entropy settings, making the evaluation somewhat less robust for these specific scenarios.
- Although the authors state that the use of multiple models and a large sample size provides a strong basis for statistical significance, the direct inclusion of explicit error bars (e.g., in tables or figures) would provide a more immediate and standard visualization of statistical variability and reinforce the claims .

---

> ### Author Rebuttal · Authors · 2025-07-29
>
> Thank you for your professional and constructive review. We appreciate your recognition of the strengths of our paper and your thoughtful suggestions. Below are our responses to your specific points:
>
> ---
>
> ### **W1. A more straightforward application of GoF tests**
>
> Thank you for this valuable observation.
>
> Indeed, **no single GoF test consistently dominates across all scenarios**—a fact we highlighted in Table 4. This diversity reflects the **classical trade-offs in hypothesis testing**: some tests are more sensitive to discrepancies in the tails (e.g., Anderson–Darling), while others emphasize central deviations (e.g., Watson’s test). Rather than prescribing a single solution, our goal in this paper is to **empirically reveal these nuanced differences** and encourage the use of GoF tests as a **flexible toolkit** for watermark detection.
>
> That said, we agree that providing more actionable recommendations is valuable. **We have summarized which GoF tests perform well under specific conditions** (e.g., low vs. high temperature, common vs. targeted edits) in Table 4. We also view **designing adaptive selection strategies**—that choose tests based on input characteristics—as an **important and promising direction** for future work.
>
> ---
>
> ### **W2. Low-quality output in low-temperature**
>
> Thank you for raising this concern.
>
> As discussed in Appendix D.2, outputs like “I; 1;1;1;1;1...” are extremely rare, even under low temperatures. These cases reflect pathological behaviors of the model and were kept for completeness. However, they account for a negligible fraction of the dataset and do not significantly influence overall detection performance in low-entropy settings.
>
> ---
>
> ### **W3. Statistical significance**
>
> We appreciate your suggestion.
>
> We understand the value of including error bars to better visualize statistical variability. In our case, each detection outcome is binary (watermarked or not), so the Type II error is computed as the empirical mean over 1,000 i.i.d. Bernoulli trials. This setting allows us to estimate the standard deviation as $\sqrt{p(1-p)/N}$, where $p$ is the observed error rate and $N$ is the number of samples. Hence, all relevant statistical uncertainty is already captured by this mean, and we chose to report just the Type II error rate for clarity. We can include binomial confidence intervals in future versions if helpful.
>
> We speculate the reviewer might be referring to **variance across multiple runs** (e.g., rerunning the entire experiment multiple times). However, this would be computationally expensive, and given our large sample size, the estimated error rates are already statistically stable.
>
> ---
>
> ### **Q1. How expensive are the GoF tests?**
>
> Thank you for raising this question. The **most expensive cost** for goodness-of-fit (GoF) tests is *calculating the critical value*.
>
> Most GoF tests—including all those used in this work—**lack non-asymptotic distributions** but do have asymptotic ones. GoF tests with **simpler asymptotic distributions**—specifically `Ney` and `Chi`—can **directly use asymptotic results** to calculate critical values. As a result, their runtime is **comparable to sum-based baselines**.
>
> However, several tests (e.g., `Kui`, `Kol`, `And`, `Cra`, `Wat`) **do not use their asymptotic distributions** for critical value computation (even though they have one) due to their complex analytical forms (e.g., the Anderson-Darling distribution [2, 3]). Instead, they rely on **Monte Carlo simulation** with a large number of samples (we use $10^5$) to estimate the critical value, which increases computational cost.
>
> Below is the time cost for detecting 100 pivotal sequences with $n = 400$ and simulation size $10^5$. For comparison, the `Ars` baseline takes $1.30 \times 10^{-3}$ seconds. As shown in the table, GoF tests that rely on Monte Carlo simulation (`Kui`, `Kol`, `And`, `Cra`, `Wat`) take roughly $10^3$ times longer to run compared to the baseline. This is mainly due to the $10^3$-fold larger simulation size required for estimating critical values.
>
> | **Test** |  **Time**   | **Time (without calculating critical value)** |
> | :------: | :-------------------: | :-----------------------------------------------------: |
> |  `Phi`   | $2.11 \times 10^{0}$  |                  $2.20 \times 10^{-3}$                  |
> |  `Kui`   | $2.03 \times 10^{0}$  |                  $1.50 \times 10^{-3}$                  |
> |  `Kol`   | $1.95 \times 10^{0}$  |                  $1.40 \times 10^{-3}$                  |
> |  `And`   | $2.61 \times 10^{0}$  |                  $4.30 \times 10^{-3}$                   |
> |  `Cra`   | $2.37 \times 10^{0}$  |                  $1.70 \times 10^{-3}$                   |
> |  `Wat`   | $2.38 \times 10^{0}$  |                  $1.80 \times 10^{-3}$                   |
> |  `Ney`   | $4.00 \times 10^{-3}$ |                           same as left                            |
> |  `Chi`   | $3.90 \times 10^{-3}$ |                           same as left                            |
>
> We will include this table in the Appendix.
>
> ---
>
> ### **Q2. Comparison with other statistical methods**
>
> **This work addresses a different problem and is not directly comparable to ours.** [1] focuses on detecting the *presence and type* of watermarking schemes in a black-box setting—without access to input parameters or watermark keys. This aligns more with an adversarial goal: identifying whether a model is watermarked and which scheme it uses.
>
> In contrast, our work assumes the watermark scheme and secret keys are known, and focuses on detecting whether a given text sequence is watermarked—this reflects the use case of LLM APIs like Gemini.
>
> That said, [1] is a very interesting and valuable contribution to the broader watermarking literature. We will mention it and discuss the difference in our next revision.
>
> ---
>
> ### **References**
>
> [1] Gloaguen, Thibaud, et al. "Black-box detection of language model watermarks." *arXiv preprint arXiv:2405.20777* (2024).
>
> [2] Anderson, Theodore W., and Donald A. Darling. "Asymptotic theory of certain" goodness of fit" criteria based on stochastic processes." *The annals of mathematical statistics* (1952): 193-212.
>
> [3] Marsaglia, George, and John Marsaglia. "Evaluating the anderson-darling distribution." *Journal of statistical software* 9 (2004): 1-5.

---

> > ### Comment · Reviewer_jT6m · 2025-08-04
> >
> > Thanks for the response. It satisfactorily addresses most of my concerns.
> >
> > The authors clarify that different GoF tests excel in different conditions and frame their contribution as providing a toolkit rather than a single prescriptive method, which I find reasonable. Their explanation regarding rare low-quality outputs at low temperatures and the minimal impact on results is convincing, and the discussion on statistical variability (binary outcomes aggregated over large samples) is also acceptable. The added detail on computational costs and the scope distinction from related black-box work resolves previous ambiguities.
> >
> > While adaptive test selection and more explicit runtime reporting remain promising future directions, these do not undermine the core contribution. The paper offers a novel and thorough empirical study of classical GoF tests for watermark detection, yielding robust insights into performance across watermarking schemes, temperatures, and edits. Just because of the significance of the study and the clarity in writing and well structured experiments, I would be increasing my scores.

---

> > > ### Author Response · Authors · 2025-08-05
> > >
> > > Thank you for your thoughtful feedback and for recognizing the value of our work. We appreciate your positive assessment and are glad that our clarifications addressed your concerns. We agree that adaptive test selection and more detailed runtime analysis are promising directions, and we will consider incorporating them in future work. Thank you again for your encouraging comments and for increasing your score.

---

### Official Review · Reviewer_qCSQ · 2025-06-30

**Clarity:** 3
**Significance:** 2
**Originality:** 3
**Rating:** 5
**Confidence:** 4

**Summary:**

In contrast to most recent work on watermarking for LLMs, this paper explores the application of goodness-of-fit testing to the detection of the signals produced by existing watermarking techniques. They survey a selection of different GoF tests and and implement them for a set of popular watermarking schemes. They empirically evaluate the efficacy of the proposed tests and compare their power in detecting watermarked texts in benign and attacked settings (edits and corruption). They systematize their observed trends in when certain GoF tests outperform the baseline pivotal statistics and present explanations as to why GoF tests might be more robust and suitable to usecases.

**Questions:**

### Questions

1. Primary request would concern the experimental setup in terms of model and prompt selection. However, it is unlikely the authors can redo significant parts of the results with modern models during the review period. As a stopgap, can it be made ultra clear what models and prompts are used for every experimental result? For some of the tables and figures clear information is provided, but not all.

2. Some experiments present token count as a parameter. Was any analysis done below n=200 or 100 tokens? Most watermarks are expected to perform well at a couple hundred tokens and concerns often center much shorter sequences of  just a few sentences or a short code snippet. Are there any experiments at shorter sequence lengths? This is potentially an extreme but practical area where GoF tests stand to make significant progress (due to the poor performance of existing techniques in that area)

3. Could more details be provided in the paper about the GoF tests for each watermarking method? Table 1 briefly notes the deviation expressions and Appendix A provides nice technical explications of each watermarking method, but these are left unconnected and so there is a gap in how the actual GoF measures are computed for each watermark (I suppose currently a reader must dig through the code to go look at the test implementations and try and reason through how they relate to each watermarking method's raw scores)



### Additional discussion regarding primary weakness:

Some of the technical points about text repetition are particularly pronounced in old small models like the OPT series. Llama 3 8B is mentioned as being included in the average for Table 2, but it appears that the small OPT model might have been used for Figures 1,2,3.

While presented well, the section on "The influence of repetition on top probabilities" discusses the tendency towards repetition in next token prediction language models as if this is a novel or understudied phenomenon.
The appendix section on related work also supports that interpretation about the authors' perspective: "Notably, we are the first to analyze the role of text repetition in watermark detection" (L580)

It is neither a new phenomenon in the general study of neural language models/LLMs nor in the study of watermarks [1]. Multiple studies discuss and experiment with methods to address the miscalibration of p-values derived from tests over pivotal statistics that rely on i.i.d. assumptions that are violated by repetition within a text [2,3]. Counting of only unique observations is one of the established fixes discussed in many papers and applied in implementations [4].

Much of the last 3 years of work on LLMs has been done to directly address issues like low entropy and degenerate behaviors like repetition. These factors both heavily interact with watermarks. Thus, this analysis would be much more valuable if performed using generations from modern models like Llama 3.1 or  3.2, Gemma 2 or 3 and Qwen 2.5 or 3, and on prompt datasets like modern long form reasoning benchmarks where models generate CoTs as this is more representative of deployment today. Rambly completions from OPT 1.7B of random C4 webtext pages are not representative of modern usage. That was a setting used in some of the earliest work in the modern wave of watermarking for LLMs but ChatGPT had just been released in 2022 and OPT was one of the only useable open source models available at that time.

In summary, the presentation of certain parts of the analysis, as well as choices in experimental setup corroborate a lack of understanding about how important the quality of the LLM used and the diversity of the text distribution are in practical takeaways such as "which GoF test should I deploy in my inference setting?"

[1] "The curious case of neural text degeneration" https://arxiv.org/abs/1904.09751 (general, repetition in weak LLMs)

[2] "Three Bricks to Consolidate Watermarks for Large Language Models" https://arxiv.org/abs/2308.00113 (Section III, empirical CDF deviation, repetition in watermark detection)

[3] "On the reliability of watermarks for large language models" https://arxiv.org/abs/2306.04634 (Appendix A.3, empirical CDF deviation)

[4] https://huggingface.co/docs/transformers/v4.52.3/en/internal/generation_utils#transformers.WatermarkDetector.ignore_repeated_ngrams (implementation of repetition corrections in standard codebases)

**Ethical Concerns:**

["NO or VERY MINOR ethics concerns only"]

**Final Justification:**

See responses.

**Limitations:**

- Primary weakness limits the ability to to apply their findings. Just as a concrete example, one might ask "Based on the experiments and summary (Table 4), in practice, which GoF test should be used for a reasoning intensive usecases such as the QwQ model performing on coding and mathematical reasoning tasks for a user?" This paper does not quite provide evidence that is relevant enough such that prescriptions could be readily applied in the real world.

**Quality:**

3

**Strengths And Weaknesses:**

### Strengths

1. Beyond the primary contemporary citation (Li et al.) their work is one of the first to examine how to improve the detection of existing watermarking methods simply by exploring the space of possible statistical tests one could use without modifying the embedding scheme.

2. They present the GoF tests and their application to selected watermarking schemes in a unified formalism and provide motivation as to why GoF tests should offer advantages over pivotal statistics in certain settings

3. Clearly presented experiments covering both modifications to sampling temperature, text length, and a variety of edit/corruption attacks.


### Weaknesses

1. Models and data used in experiments are old. While this is normally a filler criticism in reviews, because of how intertwined watermark embedding and detection performance is with the generated text distribution---a function of both the prompts and the capabilities of the model---this is actually a relatively major weak point of the empirical results.
2. The work appears to have overlooked certain other studies' prior treatment of statistical soundness issues with existing watermarking schemes and presents some known results as novel.


(...and the implementation design choice issues for empirical results actually appear related to perspective issues wrt prior work. See details in Questions section.)

---

> ### Author Rebuttal · Authors · 2025-07-29
>
> Thank you for your thoughtful and detailed review. Below, we respond to your comments and questions point by point.
>
> ---
>
> ### **1. The choice of LLMs**
>
> We appreciate your concern regarding model selection.
>
> We selected the OPT models primarily to ensure comparability with prior work \[1–4], where they have been widely used. To examine the effect of model size, we included both OPT-1.3B and OPT-13B. Acknowledging that the OPT series is relatively outdated, we also included LLaMA 3.1-8B—a significantly stronger, modern model—to evaluate whether newer architectures materially affect watermark detection.
>
> Our findings show that **differences across models have minimal impact on the relative performance of detection methods**. Instead, the **entropy of the next-token probability (NTP) distribution**—which governs how strongly watermark signals appear—plays a much more critical role. Since NTP entropy can be directly manipulated via temperature, we can effectively explore a wide range of detection regimes using temperature scaling, even on older models.
>
> In this sense, using a newer model that naturally produces higher-entropy outputs is functionally similar to evaluating an older model at a higher temperature. As demonstrated in Appendix Figures 4–6, the **performance trends across GoF tests remain consistent across models**, reinforcing our claim that entropy, not model architecture, is the primary driver of detection difficulty in this setting.
>
> ---
>
> ### **2. Experiment clarification**
>
> Thanks for pointing this out.
>
> In Appendix C.1, we provided the prompts used for text generation.
>
> All three models were included in every detection task:
>
> - Type I error on human data and Type II error evaluations (Table 2, 8; Figure 4, 5, 6)
> - Type II error under various editing types (Table 3, 9)
>
> For ablation studies:
>
> - Results are averaged across the three models for repetition rate (left parts of Figures 2, 7, 8) and for changes in Type II error after removing repeated pivotals (Figure 3).
> - In Figure 1, confidence intervals are based on pivotal sequences from all three models.
>
> For illustrative purposes:
>
> * In Figure 1, the three curves (light blue, dark blue, and orange) come from OPT-1.3B. This figure is meant to help readers intuitively understand the effect of repetition on the empirical CDF.
> * In Figures 2, 7, and 8 (right parts), the top probability distributions are shown for OPT-1.3B. Although we ran the same experiments on OPT-13B and LLaMA 3.1, the patterns were nearly identical, so we omitted those plots for brevity.
>
> Thanks for pointing this out—we'll include this clarification in the revision.
>
> ---
>
> ### **3. Shorter text sequence**
>
> This is an excellent point—thank you for highlighting it.
>
> **We did a brief analysis on shorter sequences.** In Figure 1, we present the 95% confidence intervals of the empirical CDF for both long sequences ($n=400$) and short sequences ($n=100$). The figure shows that under high temperature ($T=1$), the empirical CDF diverges noticeably from the null CDF even when $n=100$. Also, Appendix Figures 4, 5, and 6 show how the Type II error decreases as the watermark sequence length increases ($n\in[1,400]$), with particularly rapid decay at $T=1$ and $T=0.7$. These results suggest that **GoF tests can effectively detect watermark signals with relatively short sequences under high temperature**.
>
> However, we didn’t conduct a more detailed discussion on varying sequence lengths in the main text, as our focus is on demonstrating the broader potential of GoF tests as a general framework for improving watermark detection.
>
> Thanks again for the thoughtful question—this is a promising direction for future work.
>
> ---
>
> ### **4. More details about the connection between GoF and watermark**
>
> Thanks for pointing out this concern.
>
> We will certainly include more details in the Appendix. Below is a brief explanation to help avoid any potential misunderstandings.
>
> **• What is '$Y_{(t)}$' used in GoF tests?**
>
> $Y_t$ denotes the pivotal statistic (defined in L28) for the $t$-th token. Given a sequence $Y_1, \ldots, Y_n$ of pivotal statistics, $Y_{(1)} \leq Y_{(2)} \leq \dots \leq Y_{(n)}$ are their *order statistics*. For a fixed watermarking scheme, the distribution of $Y_t$ under the null hypothesis (i.e., human-written text) is known and denoted as $\mu_0$. Note that $\mu_0$ varies depending on the specific watermarking method—details are provided in Appendix A.
>
> **• Connection to GoF tests**
>
> Since we know that under the null hypothesis the pivotal statistics should always follow the known distribution $\mu_0$ independently, we can simply test whether a given sequence of $Y_1, \ldots, Y_n$ follows $\mu_0$. If not, we reject the null hypothesis and infer that the text is no way human-written and thus LLM-generated. To conduct this hypothesis testing, we use GoF tests—classical statistical tools for evaluating whether a sequence of i.i.d. samples comes from a specified distribution. More details are provided in Section 3: *Rationale for Using GoF Tests*.
>
> **• How are GoF test statistics calculated?**
>
> Each GoF test can be viewed as a function of the pivotal statistics \$Y\_1, \ldots, Y\_n\$, designed to measure the *distance* between their empirical distribution and the null distribution \$\mu\_0\$. For example, consider the Kolmogorov–Smirnov (`Kol`) test applied to the Gumbel-max watermark with a significance level of \$\alpha = 10^{-2}\$:
>
> 1. Compute the pivotal statistics: $Y_1, \ldots, Y_n$
> 2. Sort them in ascending order: $Y_{(1)} \leq \dots \leq Y_{(n)}$
> 3. Compute the KS test statistic: $D_n = \max_{1 \leq i \leq n} (\max( Y_{(i)} - \frac{i - 1}{n}, \frac{i}{n} - Y_{(i)} ))$
> 4. Use Monte Carlo simulation to estimate the critical value $\gamma$ corresponding to the chosen $\alpha$
> 5. If $D_n > \gamma$, reject the null hypothesis; otherwise, accept it.
>
> More details for other tests are provided in Section 3: *Introduction of GoF tests*.
>
> ---
>
> ### **5. Reiteration of known results (repeated n-gram problem)**
>
> Thanks for pointing this out.
>
> **Our work approaches the repetition issue from a different perspective**.
>
> First, we acknowledge that repetition in LLM outputs has been identified and analyzed in prior works, as we clearly state in the paper (L264). We'll also add [1] to the citations to better reflect this.
>
> Second, we appreciate that [2, 3] addressed and effectively handled the repeated text issue in watermark detection. However, rather than focusing on how to deal with repeated text, we analyze *why repetition makes sum-based methods less effective*—see ‘The influence of repetition on top probabilities’. We further explore *how repetition shifts the **empirical CDF of pivotal statistics** (not p-values) and impacts GoF detection*—see ‘The influence of repetition on the empirical CDF’. This analysis brings out a key advantage of GoF: it’s particularly effective at capturing deviations between the empirical CDF of pivotal values and the null $\mu_0$.
>
> That said, we agree that the current wording in the paper could cause confusion. We will revise it to clarify that we analyze the role of repetition in watermark detection *from a new perspective—by examining its impact on the statistical behavior of pivotal values and the reliability of GoF-based detection, rather than proposing fixes or pre-processing heuristics*.
>
> ---
>
> ### **References**
>
> [1] Li, Xiang, et al. "Robust detection of watermarks for large language models under human edits." arXiv preprint arXiv:2411.13868 (2024).
>
> [2] Kuditipudi, Rohith, et al. "Robust distortion-free watermarks for language models." arXiv preprint arXiv:2307.15593 (2023).
>
> [3] Li, Xiang, et al. "A statistical framework of watermarks for large language models: Pivot, detection efficiency and optimal rules." The Annals of Statistics 53.1 (2025): 322-351.
>
> [4] Kirchenbauer, John, et al. "A watermark for large language models." International Conference on Machine Learning. PMLR, 2023.

---

> > ### Comment · Reviewer_qCSQ · 2025-08-06
> > **Response**
> >
> > The reviewer appreciates the thorough responses to the points raised in the original review. While there are a few remaining comments to be made the rebuttal has generally increased the reviewer's confidence in the work.
> >
> > ### 1.
> >
> > The reviewer is quite aware of the fact that prior work has used OPT models, but for a paper submitted in 2025, this is needlessly out of date; model quality has improved drastically since then, including small models for researchers with limited compute resources. While, the technical point made about entropy is of course correct, and the results suggest general consistency among the models that were tested, this does not discount or refute the reviewers continued assertion that experimenting with stronger models with different next token distributions is not equivalent to applying simple temperature scaling.
> >
> > Temperature scaling is a coarse grained way to change the overall entropy of the NTP distribution that has an existing dispersion, but watermark embedability in practice is influenced by the "flexibility" of the generative models distribution in response to various prompts. A model that can respond in k equally high quality ways to any one prompt is more likely to be readily watermarkable (significant test statistics, minimal impact on text quality) than a model that can only produce a single response to the prompt (any deviation sufficient to cause significant statistics would cause text degradation). For this reason, watermarks that operate as decoders on the NTP distribution harness the availability of multiple _plausible_ (top k) token options at each step. Overall distribution entropy is overly influenced on the long tail of unlikely tokens in the vocabulary, none of which factor into the realized choices under a watermark that predominantly respects the original sampling distribution, which is essentially all useable watermarks. Except in the limit of extreme temperatures (low or high) standard shannon entropy and temperature scaling alone do not actually characterize or modulate the NTP distribution in the same way as the reviewer is describing. The training data, weights, architecture and quality of the model overwhelmingly determine the distribution' structural properties that might make a LLM more or less watermarkable, or might modify the ability of a specific test to detect the embedded watermark.
> >
> > While the authors point to Fig 6 in the appendix as evidence for the generalization they assert about entropy over different models as being the primary factor, it is worth remarking that while the trend as a function of temperature is similar across models, row 4 and row 6 are characteristically more similar than row 5, despite the parameter count of 5 being between the two and this is likely because of training data and architectural similarities between the OPTs versus the vastly different Llama 3.1. It is fine to claim that, in the experimental settings considered, "performance trends across GoF tests remain consistent across models". This is an accurate statement if considered within the scope of the results shown. However, the reviewer is quite certain that if evaluated across both diverse classes of LLMs and text domains the reviewers would find more variation in test performance across different GoF test flavors than their study currently suggests; perhaps, some more interesting and systematic trends as a function of test type would even be uncovered.
> >
> > As a result, the reviewer is not sure that the work will attract much practical interest _especially_ given the lack of clear prescriptions provided for pairing tests and usecases. In the end, this is why the review critique remains warranted. If the experiments as they were performed, yielded very clear prescriptions, then maybe this would not matter, but "lack of effect" is a difficult result to act on or otherwise try and generalize. Essentially, future work will have to re-perform relevant parts of the entire study on their own models of interest on prompts reflective of real usage to begin to discern actionable insights as to what tests to use or whether any benefit is yielded over sum base tests.

---

> > ### Comment · Reviewer_qCSQ · 2025-08-06
> > **Response**
> >
> > ### 2.
> >
> > Thank you for the clarifications. If the authors can include these single phrase additions in captions and other prose sections where they were missing, this would increase the clarity of the results for the reader.
> >
> > ### 3.
> >
> > The authors and reviewer seem to have an accord.
> >
> > ### 4.
> >
> > Thank you for the added clarity. The reviewer believes that working the enumerated generalized pseudo-algorithm as to how the GoF tests are performed wrt each type of watermark statistic is quite useful and will add clarity if added at the end of Sec 3. perhaps after the additional details itemization and before the heading for Sec 4.
> >
> > ### 5.
> >
> > Understood. Please make the distinction between the treatment of the effect of non-iid statistics caused by repetition on both sum and GoF methods more clear in the main work; this will both treat the literature better and highlight the novelty of your analysis.
> >
> > As an additional connection between this section on repetition and the primary detractor of the work discussed above, one notable case in point is that a reasonable amount of space in the work is devoted to the treatment of repeated text. This is _only_ a practical issue for very weak, small, old models (or at best, modern base models with no instruction tuning, let alone RL posttraining) and so, this analysis, while sound, is of questionable value to the community since the models it studies are not what would ever be used in practice.
> >
> > ### Verdict
> >
> > As stated above, the rebuttal has generally improved the reviewer's appraisal of the work. Their opinion that the empirical choices made do a disservice to the methodological contributions of the work regarding the study of GoF tests remains. But, the reviewer is willing to increase their score to a weak accept trusting that the authors will do their best to incorporate the suggestions made by all reviewers.

---

> ### Author Response · Authors · 2025-08-06
> **Thanks for your feedback**
>
> Thank you for your thoughtful feedback. We are glad our clarifications improved your appraisal. The followings are our response to your comment.
>
> ### **1. The choice of LLMs**
>
> Thanks for continuing to discuss this with us. We respectfully offer the following clarifications:
>
> * **High watermarkablility makes detection easier so less focus on it.** We agree with your point that temperature scaling cannot fully control or replicate the structure of the NTP distribution. As you noted, stronger models often produce multiple equally plausible next tokens, which makes them inherently more watermarkable. In such cases, detection is often **easier** than simply increasing the temperature of a weaker model.
>
>   **We acknowledge this distinction.** However, in these easier cases, most detection methods—including ours—already achieve near-perfect performance. The performance on more watermarkable models is expected to be (and should be) even better. For this reason, the research community, including us, tends to focus on the **more challenging scenarios**, such as when the NTP distribution is peaked and exhibits relatively **low entropy**. This might explain why most watermarking studies—including ours—do not pursue the most recent LLMs, since the **core difficulty lies in detecting watermarks under less unfavorable conditions**, NOT in easy-to-detect cases. For example, [5] conducts experiments using the Gemma-Instruct models [6], which exhibit much lower-entropy NTP distributions compared to base Gemma models.
>
>   That said, incorporating more recent models can certainly enhance the study and provide additional perspective, even if not strictly necessary. While the community often focuses on challenging detection settings, adding newer models could offer complementary insights. We will include this point in the limitations section (Section 5).
>
> * **Practical utility and flexibility of GoF tests.** Our main contribution is introducing genera GoF tests to enrich the watermark detection toolkit. While we acknowledge the diversity of LLMs and text domains—and that GoF test performance may vary across different real-world cases—we emphasize that the **cost of applying GoF tests is very low**. As detailed in our response ([Q1](https://openreview.net/forum?id=YES7VDXPV8&noteId=6j3K8OAOY6)) to Reviewer jT6m, these tests require only lightweight statistical computations on CPUs and can be executed in parallel. Given this low overhead, users can run all suggested GoF tests and select the best-performing one in just a few minutes (depending on the time for computing critical values). To further support adoption, we will release well-organized Python code to make these tests easy to use in practice (note that we already attached them in this submission).
>
> ---
>
> ### **References**
>
> [5] Dathathri, Sumanth, et al. "Scalable watermarking for identifying large language model outputs." Nature 634.8035 (2024): 818-823.
>
> [6] Team, Gemma, et al. "Gemma: Open models based on gemini research and technology." *arXiv preprint arXiv:2403.08295* (2024).

---

> ### Author Response · Authors · 2025-08-06
> **Thanks for your feedback**
>
> ### **2. Plan for the next revision**
>
> Thank you for the helpful suggestions. Based on your feedback, we outline our revision plan as follows:
>
> * **Clarify the scope of model generalization claims.** We will revise the statement about consistency across models to explicitly state that the observed performance trends hold within the scope of the models tested. We will also acknowledge that differences in model architecture and training (e.g., between OPT and LLaMA 3.1) may influence detection outcomes. Additionally, we will clearly inform readers that, given the diversity of LLMs and text domains, *greater variation* in GoF test performance may exist than our current study suggests.
>
> * **Add clarifying phrases to captions and prose.** We will insert concise clarifications in figure captions and relevant text to improve clarity and avoid ambiguity in the presentation of results.
> * **Add a pseudo-algorithm for applying GoF tests.** We will include an enumerated, generalized pseudo-algorithm for implementing the GoF tests at the end of Section 3 (space permitting; otherwise, it will be added to the Appendix) to help readers better understand the implementation flow.
> * **Clarify novelty in how repetition affects GoF vs sum-based methods.** We will strengthen our discussion of how repetition impacts both types of methods and emphasize our **unique analytical perspective**—*specifically, by examining how repetition influences the statistical behavior of pivotal values and the reliability of GoF-based detection, rather than focusing on mitigation or preprocessing strategies*. We will also clearly frame our repetition analysis as an edge-case study and point out that repetition is less prevalent in more recent LLMs.
> * **Further discussion on the influence of different LLMs.** We will revise and expand the discussion in Appendix D.1 to include our rationale for the choice of models, acknowledge the limitations of not including newer models, and clarify the implications for generalizability. This should offer readers clearer guidance on interpreting our results in relation to model diversity.
>
> Finally, thank you again for the thoughtful and constructive discussion, for your willingness to raise the score, and for your trust in us to thoughtfully incorporate the suggestions from all reviewers—your feedback has been truly appreciated.

---

> > ### Comment · Reviewer_qCSQ · 2025-08-06
> >
> > Some final soapbox points:
> >
> > - The fact that "low entropy scenarios are challenging" is only a useful distinction, and one the community should pursue, if the setting being discussed is practical. Code generation or mathematical reasoning are useful low entropy settings to study but the low entropy NTP distributions of terrible language models are not an interesting setting to study. Why? because the reason that the latter class is low entropy is structurally different from the former, and so it's not clear the experiments in the latter case will not actually inform the former case even though the summary statistic is correlated. But I truly digress.
> >
> > - Note that whatever Google chooses to evaluate in a Nature paper is not a very reliable datapoint; many motivations play into this considering the other, potentially much more expressive models (and instruct/reasoning variants) at their disposal. It has long been an interest of this reviewer to see watermarking experiments performed on GPT, Gemini, and Claude class models. Even the earliest papers in the modern "GenAI watermarking" era discuss how additional entropy helps watermarkability and therefore highly capable frontier systems might actually be the most amenable to the technique. Alas, academics cannot answer these questions in open research.
> >
> > The revision plan is appreciated. I hope to see the paper presented at the conference!

---

> > > ### Author Response · Authors · 2025-08-07
> > > **Appreciation for thoughtful insights and encouragement**
> > >
> > > Thanks for sharing the thoughtful reflections--we really appreciate your perspective.
> > >
> > > We fully agree that not all low-entropy settings are equally meaningful, and we acknowledge the important distinction between *task-driven* low entropy (e.g., in code generation or math reasoning) and *model-induced* low entropy from weaker models. A natural future direction for our work is to focus more on practical use cases, such as mathematical reasoning, using stronger LLMs and pairing them with theoretical guarantees.
> > >
> > > We also appreciate your insight on Google's work. We also share your interest in watermarking on frontier models like GPT, Claude, and Gemini, and we agree these represent the most promising ground for future study. On that note, Google has announced plans to deploy their watermark (SynthID) in Gemini ([link](https://deepmind.google/discover/blog/watermarking-ai-generated-text-and-video-with-synthid/)), which may hopefully offer new insights into watermark development in real-world systems.
> > >
> > > Thank you again for your support and encouragement—we’re honored by your interest and hope the revised paper meets your expectations.

---

### Official Review · Reviewer_aMbH · 2025-07-02

**Clarity:** 3
**Significance:** 2
**Originality:** 2
**Rating:** 4
**Confidence:** 5

**Summary:**

In this paper, the authors propose an empirical overview of classic Goodness of Fit tests -- i.e non-parametric test in a simple, binary hypothesis test setting -- applied to LLM watermarking. The main idea is to replace the usual detection rule of LLM watermarks (classically a sum of on statistics computed from a token, a key and hash computed on previous tokens) with such goodness-of-fit tests and observe if it does lead to improvements compared to the naive detection rules.
Experimentally, the authors study:
- 8 GoF tests
- 3 "distortion-free" watermarking schemes
- 3 LLMs (and 4 temperature settings for each)
The study highlights two relevant settings of interest:
- low-temperature (i.e almost deterministic) generation
- common text edits -- what I would call robustness evaluation --  and information-rich edits -- what i would call adversarial evaluation.


Through these experiments the authors claim that GoF tests outperform classic methods in robustness even at low temperatures.

**Questions:**

1. Could the authors improve their evaluation by 1) working at low FPR (at least $10^{-6}$ ) and 2) use either non-asymptotic  $p$-values  **or**  do a precise empirical evaluations of the theoretical FPR vs empirical FPR (see Figure 2 in [1] or Figure 4 [2] for good examples)?
2. Since the repeated n-gram problem was extensively treated in [1], maybe simply use their hashing function as described in Section III.C ? If using the non-asymptotic p-value, this should guarantee a probability of false alarm for both Aaronson and KGW. This would allow a fair comparison between the author's method and other schemes.
3. Why were these LLMs chosen? I see two OPT models at different number of parameters (very old models) and the more recent Llama-3.1. I see no presented rationale for this choice (Appendix D is not very enlightening in this regard) and would appreciate one. Even better yet, given that the authors are interested in the impact of the entropy during generation (as controlled by the temperature), why not provide the distribution of the entropy for each model computed for each step and each texts?
4. *(Minor)* What is the length of texts generated? This has a (very!) substantial impact on detection power and should be controlled accordingly.


Regarding low FPR, I am aware it is not achievable for some algorithms (notably Inverse Transform), I believe it is fine to either forego them completely (the absence of a sound p-value/FPR guarantees means a watermarking scheme is essentially useless) or present the results separately for a higher FPR.

**Score increase or decrease** : Given the current state of the paper, I see no reason to decrease the score. On the other hand, I am ready to increase the score provided my gripes around the methodology are addressed.

**Ethical Concerns:**

["NO or VERY MINOR ethics concerns only"]

**Final Justification:**

The authors made their case for the usefulness of their method at low temperature regime.
Even though I'm still divided on the concrete application of asymptotic methods when non-asymptotic ones are available, it is now more a question of philosophy and methodology: the authors claim that simulation can be sufficient (which is true though quite costly).
Other questions were answered adequately.

I am not entirely sold on the work due to its asymptotic nature and lack of significance power increase in moderate temperature regime, but given the efforts made by the authors, it would be unfair not to raise my score, hence me bumping it to a borderline accept.

**Limitations:**

Nothing to report.

**Quality:**

3

**Strengths And Weaknesses:**

## Qualities

- **Clarity and scope**: The paper is overall very well written and clear in what it is trying to achieve -- an empirical evaluation of the GoF tests  in the setting of LLM watermarks. The scope is well respected. Given the informations provided in the paper, I am confident I should be able to reproduce the results.

- **Experimental design**: The experiments are performed on diverse datasets, several LLMs (though I will question the specifics), and several watermarking schemes and detection methods. I am also quite happy to see that the authors make a difference between adversarial and non-adversarial edits, the distinction between robustness and security still being quite blurry in the ML community.


## Weakness

Before listing the weakness, I have to say that they are not purely the fault of the authors. From what I am seeing, the authors have been following what has more or less become the standard in evaluating watermarking algorithms in the ML community ; they cannot be blamed for this. Nevertheless, these standards are flawed when compared to those which were enforced in the classical era of watermarking [4]. Seeing how watermarking is gaining traction in the ML community, I believe it is time to direct papers to more rigorous and stringent evaluations since these designs are meant to eventually be used in the real-world.

- **The $P_{FA}$ is not controlled**: Maybe the most important design aspect of watermarking is the ability to guarantee a certain level for the Type I error. The authors only provide an empirical evaluation of this error on a thousand human texts. This is **not** sufficient. What a good watermarking scheme should strive for is 1) a theoretical model of the detector statistic under $\mathcal{H}_0$. Usually, the statistic should follow a uniform distribution to allow for a simple computation of the $p$-value. And 2) an empirical validation of the model $\mathcal{H}_0$, with usually at least a million samples to allow an acceptable confidence interval between $10^{-5}$ and $10^{-6}$ . See [1] or [2] for recent examples of work performing such evaluation correctly (though they do lack a visible confidence interval!).
-  **$P_{FA}$ is too high**: Even assuming the authors correctly control their Type I error, it is far too high. The authors fix their significance level to $\alpha=0.01$ and claim it to be a standard choice in previous work (with many references to back it up). From what i can gather, this standard seems to be a bad consequence of the fact that [3] was not able to provide theoretical guarantees for their schemes, leading the authors of [3] to resort to empirical validation. Due to its high computational cost, $\alpha=0.01$ was the best value in reach. Current state-of-the-art schemes are already able (and should!) to reach excellent power at far lower Type I error -- see the appendix A.X of [2] for proofs of this fact. A usual value in the classical era of watermarking was $\alpha=10^{-6}$, which given the high number of LLM generated texts per does not even seem that low for practical deployment in the real-world.

Unrelated weaknesses include:
- **Reiteration of known results**: The whole section of the "missing factor at low temperatures" was extensively treated in [1]. Furthermore, the work actually proposed new hashing and detection methodology to completely solve the problem -- keeping guarantees in terms of Type I error at the cost of lower power.
- **Lack of significance/usefulness**: In its current state, the paper does not provide a model of the $p$-values -- i.e the statistics under $\mathcal{H}_0$ -- limiting the usefulness of the GoF method in practice since it prevents strong guarantees in terms of Type I errors. This is especially a problem since strong guarantees can be obtained for e.g. KGW and Aaronson's scheme by using non-asymptotic detector [1].


[1] Fernandez, P., Chaffin, A., Tit, K., Chappelier, V., & Furon, T. (2023, December). Three bricks to consolidate watermarks for large language models. In 2023 IEEE International Workshop on Information Forensics and Security (WIFS) (pp. 1-6). IEEE.

[2] Giboulot, E., & Furon, T. (2024, December). WaterMax: breaking the LLM watermark detectability-robustness-quality trade-off. In NeurIPS 2024-38th Conference on Neural Information Processing Systems (pp. 1-34).

[3] Kuditipudi, R., Thickstun, J., Hashimoto, T., & Liang, P. Robust Distortion-free Watermarks for Language Models. Transactions on Machine Learning Research.

[4] Bas, P., Furon, T., Cayre, F., Doërr, G., & Mathon, B. (2016). Watermarking security: fundamentals, secure designs and attacks. Berlin: springer.

---

> ### Author Rebuttal · Authors · 2025-07-29
>
> Thank you for your professional review and valuable comments. Below are our responses, with clarifications and new evidence addressing your concerns.
>
> ---
>
> ### **Q1. Concerns about controlling Type I error**
>
> Thank you for highlighting the importance of Type I error control.
>
> * **Why we chose $\alpha = 0.01$:**
>    * First, we used 1,000 human texts for empirical evaluation. Under this sample size, \$\alpha = 0.01\$ is practical—it corresponds to 10 expected false positives, which is a reasonable estimate.
>    * Second, we chose this threshold for **consistency with prior influential work**. Notably, Google’s SynthID watermark \[2], reportedly ready for deployment in Gemini ([link](https://deepmind.google/discover/blog/watermarking-ai-generated-text-and-video-with-synthid/)), also adopts \$\alpha = 0.01\$.
>
> * **Can GoF tests offer better Type I error control?**
>
>   **Yes**. To further assess this, we sampled 100k texts from the C4 dataset and removed repeated pivotals, following the approach in [1]. Even at this larger scale, GoF tests still maintain strong Type I error control—for example, achieving empirical FPRs as low as $10^{-5}$. The table below shows theoretical FPR vs. empirical FPR (We also created a plot similar to Figure 2 in [1], as suggested by the reviewer, but we cannot include it here due to rebuttal constraints; it will be added to the paper.).
>
> | **Test**  | **Watermarks** | **α = 10⁻²** | **α = 10⁻³** | **α = 10⁻⁴** | **α = 10⁻⁵** |
> | :-------: | :------------: | :----------: | :----------: | :----------: | :----------: |
> | **`Phi`** |     Gumbel     |  9.25×10⁻³   |   5.8×10⁻⁴   |   2.0×10⁻⁴   |   1.0×10⁻⁵   |
> |           |   Transform    |  7.10×10⁻³   |   4.4×10⁻⁴   |   2.0×10⁻⁴   |   2.0×10⁻⁴   |
> |           |    Synthid     |  7.91×10⁻³   |  2.25×10⁻³   |   1.3×10⁻⁴   |   1.0×10⁻⁵   |
> | **`Kui`** |     Gumbel     |  9.41×10⁻³   |  1.06×10⁻³   |   2.0×10⁻⁴   |   1.0×10⁻⁵   |
> |           |   Transform    |  1.49×10⁻²   |  1.73×10⁻³   |   4.2×10⁻⁴   |   2.0×10⁻⁴   |
> |           |    Synthid     |  1.03×10⁻²   |  1.25×10⁻³   |   1.0×10⁻⁴   |   1.0×10⁻⁵   |
> | **`Kol`** |     Gumbel     |  1.03×10⁻²   |   6.0×10⁻⁴   |   1.0×10⁻⁴   |   1.0×10⁻⁵   |
> |           |   Transform    |  1.58×10⁻²   |  2.15×10⁻³   |   5.8×10⁻⁴   |   1.0×10⁻⁵   |
> |           |    Synthid     |  9.64×10⁻³   |   9.9×10⁻⁴   |   5.0×10⁻⁵   |   3.0×10⁻⁵   |
> | **`And`** |     Gumbel     |  1.03×10⁻²   |   5.9×10⁻⁴   |   1.0×10⁻⁴   |   1.0×10⁻⁵   |
> |           |   Transform    |  1.30×10⁻²   |   5.8×10⁻⁴   |   2.0×10⁻⁴   |   1.0×10⁻⁵   |
> |           |    Synthid     |  1.20×10⁻²   |  1.18×10⁻³   |   8.0×10⁻⁵   |   4.0×10⁻⁵   |
> | **`Cra`** |     Gumbel     |  9.47×10⁻³   |  1.25×10⁻³   |   1.0×10⁻⁴   |   1.0×10⁻⁵   |
> |           |   Transform    |  1.27×10⁻²   |  1.08×10⁻³   |   2.0×10⁻⁴   |   1.0×10⁻⁵   |
> |           |    Synthid     |  1.04×10⁻²   |  1.28×10⁻³   |   8.0×10⁻⁵   |   1.0×10⁻⁵   |
> | **`Wat`** |     Gumbel     |  1.08×10⁻²   |  1.07×10⁻³   |   2.0×10⁻⁴   |   1.0×10⁻⁵   |
> |           |   Transform    |  1.49×10⁻²   |  2.13×10⁻³   |   2.0×10⁻⁴   |   2.0×10⁻⁴   |
> |           |    Synthid     |  1.01×10⁻²   |   8.5×10⁻⁴   |   2.0×10⁻⁵   |   1.0×10⁻⁵   |
> | **`Ney`** |     Gumbel     |  1.12×10⁻²   |   8.7×10⁻⁴   |   2.0×10⁻⁴   |   1.0×10⁻⁵   |
> |           |   Transform    |  1.53×10⁻²   |  1.07×10⁻³   |   2.0×10⁻⁴   |   2.0×10⁻⁴   |
> |           |    Synthid     |  1.59×10⁻²   |  2.00×10⁻³   |   2.8×10⁻⁴   |   3.0×10⁻⁵   |
> | **`Chi`** |     Gumbel     |  8.18×10⁻³   |   8.6×10⁻⁴   |   1.0×10⁻⁴   |   1.0×10⁻⁵   |
> |           |   Transform    |  1.12×10⁻²   |   5.8×10⁻⁴   |   2.0×10⁻⁴   |   2.0×10⁻⁴   |
> |           |    Synthid     |  1.17×10⁻²   |  1.07×10⁻³   |   1.4×10⁻⁴   |   4.0×10⁻⁵   |
>
> * **Why GoF works in practice**:
>
>    Beyond the watermark’s design, the key factor in maintaining a low Type I error is how closely the pseudo-randomness resembles true randomness—this was also highlighted in [1]. For GoF tests, even though they lack non-asymptotic distributions, they can still offer reliable Type I error control in practice. This is because the pivotal statistics for human-written tokens behave almost like samples from a true uniform distribution after repeated values are removed. The concern about non-asymptotic distributions is further addressed in W2.
>
> ---
>
> ### **Q2. Reiteration of known results (repeated n-gram problem)**
>
> Thanks for pointing out this issue.
>
> We agree that [1] pointed out and addressed the repeated text issue in watermark detection. However, **our work studies the same issue from a different perspective**. Specifically, rather than focusing on how to deal with repeated text, we analyze **its effects on watermark detection**, such as *why repetition makes sum-based methods less effective*—see ‘The influence of repetition on top probabilities’. We also further explore *how repetition shifts the empirical CDF and impacts GoF detection* in ‘The influence of repetition on the empirical CDF’. More importantly, this analysis highlights a key strength of the GoF method: it’s particularly effective at detecting differences between the empirical CDF and the null $\mu_0$.
>
> ---
>
> ### **W2. Lack of significance/usefulness**
>
> We appreciate your concern about the absence of non-asymptotic guarantees.
>
> We acknowledge that Aaronson's score has a non-asymptotic distribution, but **a non-asymptotic distribution is NOT necessary to well-control the Type I error**. The most straightforward way to ensure valid Type I error control is to use test statistics with a known non-asymptotic distribution. Most GoF tests in the statistics literature only have known limiting distributions as $n \to \infty$ (we will include the limiting distributions of the GoF test statistics we use in the Appendix). Despite this, **the empirical results in the table above show that these tests already provide practically reliable Type I error guarantees.**
>
> It’s also worth noting that **the limiting distributions offer valuable theoretical tools, but in practice, simulations offer users a more flexible and convenient way** to compute critical values—like in Python package [SciPy](https://docs.scipy.org/doc/scipy/reference/generated/scipy.stats.goodness_of_fit.html), due to the complex forms of some distributions (e.g., the Anderson-Darling distribution [3,4]).
>
> Finally, simulation is a theoretically rigorous approach from both **theoretical and practical perspectives**. Its theoretical validity parallels that of conformal inference: for i.i.d. data points, the empirical $\alpha$-quantile of the sampled statistics serves as a valid critical value for a new test point due to the property of exchangeability. From a practical standpoint, simulation is also widely adopted in the watermarking literature—for instance, [2] uses a simulation-based method to estimate the distribution of human text (see Appendix A.4).
>
> ---
>
> ### **Q3. The choice of LLMs and analysis of the entropy**
>
> Thanks for raising this question.
>
> Many previous works used OPT models in their experiments[5-8], so we chose OPT to stay consistent for comparison. We also included two versions of OPT to examine whether model size significantly impacts GoF performance. If you think that OPT is relatively outdated, we also tested with the more recent LLaMA 3.1 to see if newer models yield noticeably different results.
>
>   Instead of reporting entropy directly, we analyze **top probabilities**, which **closely track entropy**:
> For the NTP distribution $\mathbf{P} = (P_1, ..., P_{|\mathcal{W}|})$ where $\mathcal{W}$ denotes the LLM's vocabulary, the entropy is approximated by $\operatorname{Ent}(\boldsymbol{P})=\widetilde{\Theta}\left(1-\max _w P_w\right)$ up to logarithmic factors. Figures 2, 7, and 8 present these top probabilities for repeated tokens, helping us infer entropy concentration patterns at low temperatures.
>
> ---
>
> ### **Q4. Text length**
>
> Thanks for raising this question.
>
> The length of the generated texts is fixed at 400 tokens—consistent with prior works [2, 5, 7]—across all models, temperatures, and watermarking methods, unless explicitly stated otherwise.
>
> ---
>
> ### **References**
>
> [1] Fernandez, Pierre, et al. "Three bricks to consolidate watermarks for large language models." *2023 IEEE international workshop on information forensics and security (WIFS)*. IEEE, 2023.
>
> [2] Dathathri, Sumanth, et al. "Scalable watermarking for identifying large language model outputs." Nature 634.8035 (2024): 818-823.
>
> [3] Anderson, Theodore W., and Donald A. Darling. "Asymptotic theory of certain" goodness of fit" criteria based on stochastic processes." *The annals of mathematical statistics* (1952): 193-212.
>
> [4] Marsaglia, George, and John Marsaglia. "Evaluating the anderson-darling distribution." *Journal of statistical software* 9 (2004): 1-5.
>
> [5] Li, Xiang, et al. "Robust detection of watermarks for large language models under human edits." arXiv preprint arXiv:2411.13868 (2024).
>
> [6] Kuditipudi, Rohith, et al. "Robust distortion-free watermarks for language models." arXiv preprint arXiv:2307.15593 (2023).
>
> [7] Li, Xiang, et al. "A statistical framework of watermarks for large language models: Pivot, detection efficiency and optimal rules." The Annals of Statistics 53.1 (2025): 322-351.
>
> [8] Kirchenbauer, John, et al. "A watermark for large language models." International Conference on Machine Learning. PMLR, 2023.

---

> > ### Comment · Reviewer_aMbH · 2025-08-04
> >
> > I thank the authors for taking the time to answer my questions and provide further numerical results.
> >
> > >  Even at this larger scale, GoF tests still maintain strong Type I error control—for example, achieving empirical FPRs as low as $10^{-5}$.
> >
> > I agree with the authors that the empirical FPR shown in the table is more often that not, close to the chosen level $\alpha$. However, there I can also discern several times that it is underestimated (which is serious) for the Transform algorithm at $\alpha= 10^{-5}$. This could very well be due to the fact that the confidence interval is quite bad at this regime for 100k, I will thus not argue further on this point. Nevertheless, it does illustrate my main gripe with the simulation approach: it cannot, and in this sense I respectfully disagree with the authors, provide **strong** control over the FPR. One can only bound the FPR insofar as 1) the simulation has allowed to reach an acceptable confidence interval at the chosen level and 2) the sample distribution used for the simulation is close/the same as the distribution of the sample at test time. Clearly this is difficult to reach for text. Yet, I will agree that sometimes, asymptotics is the best that the applied statistician can resort  to. I believe this is not the case for text watermarking where non-asymptotic statistics exist and are both theoretically and practically valid: the prescribed level $\alpha$ **exactly** bounds the empirical FPR. In practical, critical settings, one wants to **certify** the FPR (in court for example) and thus should not rely on non-exact statistics when possible.
> >
> > For these reasons, I see less value in using methods that necessarily rely on asymptotics.
> >
> > -> Why didn't the authors provide the TPR at these new FPR ?
> >
> >
> > I acknowledge the answers for Q1 and Q2, thank you for the clarifications.
> >
> > > The length of the generated texts is fixed at 400 tokens
> >
> > Thank you. I apologize because this information was already clearly indicated by the authors in Table 2. I now understand that indeed, the performance of the watermarking schemes are quite terrible at low temperatures, though non distortion-free schemes like KGW or Watermax should fare better in that regard. Nevertheless, some GoF tests do bring impressive performance gain in this regime. I will revise my judgement in favor of the authors for low tempratures, though I still believe that, in the high temperature regime, the presented gains are not significant (i.e. a lower FPR should have been chosen).
> >
> >
> > Under this light, I can see value in this work as a benchmark against which schemes with more theoretical guarantees should be put against in the low temperature regime. It does provide a simple yet effective comparison point. I am still not sold on its relevance in practical setting due to its asymptotic nature.
> >
> > I might increase my score slightly since my concerns have mostly been addressed.

---

> ### Author Response · Authors · 2025-08-05
> **Thanks for your feedback**
>
> Thank you for your thoughtful and constructive feedback. We're glad that our clarifications have addressed most of your concerns. As you noted, our primary point of divergence lies in the **asymptotic nature of GoF tests** and their role in practical detection systems.
>
> ---
>
> **1. Type II Error at Low FPR**
>
> Due to the character limit in the initial rebuttal, we were unable to include Type II results. We now report Type II errors at a stricter Type I error level of \$10^{-5}\$ for temperature 0.7 and sequence length \$n = 400\$, complementing Table 2. Here, “Baseline” denotes the best-performing method among all baseline detectors. Results are averaged over three LLMs. We will add the additional results in the Appendix (e.g., Type II errors at $10^{-4}$ and $10^{-3}$  Type I errors).
>
> These results demonstrate that **GoF tests retain strong detection power even under tighter FPR constraints**, especially at higher temperatures as discussed in Section 4.2.
>
> | Watermark | `Baseline` | `Phi` | `Kui` | `Kol`     | `And`     | `Cra` | `Wat` | `Ney` | `Chi`     |
> | --------- | :--------: | ----- | ----- | --------- | --------- | ----- | ----- | ----- | --------- |
> | Gumbel    |   0.011    | 0.011 | 0.007 | 0.009     | 0.007     | 0.009 | 0.012 | 0.006 | **0.004** |
> | Transform |   0.026    | 0.173 | 0.014 | **0.012** | **0.012** | 0.016 | 0.029 | 0.018 | 0.015     |
> | Synthid   |   0.042    | 0.180 | 0.046 | 0.041     | 0.039     | 0.047 | 0.118 | 0.041 | **0.027** |
>
> ---
> **2. On Bounding Type I Error in GoF Tests**
>
> We respectfully offer the following clarifications to your points about bounding FPR:
>
> * **Confidence intervals**: The accuracy of empirical critical values can be arbitrarily improved by increasing simulation size. Since the standard error of quantile estimates scales as \$1/\sqrt{n}\$, simulating more samples narrows the confidence interval and increases reliability. Importantly, this calibration can be done *before deployment*, given that the distribution of pivotal statistics is known for a given watermarking scheme.
>
> * **Match between simulation and test-time distributions**: In our setup, the pivotal statistics (e.g., for Gumbel-max) follow a known distribution (e.g., uniform on $\[0,1]\$) both in simulation and test-time. Thus, the test statistics used in GoF methods follow the same distribution in both phases. If this assumption were violated, even exact non-asymptotic methods (e.g., `Ars`) would lose their guarantees, as they also rely on this distributional assumption.
>
> Therefore, we argue that **both criteria for bounding FPR can be satisfied** in our framework.
>
> ---
>
> **3. Complementary Role of GoF Tests**
>
> We do not view GoF tests as a replacement for sum-based or non-asymptotic methods, but rather as **complementary tools**. While methods like `Ars` target high-value pivotal statistics, GoF tests are designed to detect **distributional shifts**, offering enhanced robustness especially under semantically rich or information-preserving edits (see Section 4.3). Despite relying on asymptotic or simulated critical values, GoF tests **enrich the detection toolkit** by capturing distribution-level anomalies.
>
> We agree that in high-stakes applications (e.g., legal proceedings), relying solely on asymptotic methods may be insufficient. However, we believe that **no single method should serve as the only defense**. A robust system should draw from multiple methods, each with different strengths—e.g., `Ars` for sensitivity to large deviations and GoF for distributional anomalies.
>
> To reflect this point, we will add a discussion on both the **rationale** for using GoF tests and the **potential risks** associated with their asymptotic nature, to help guide and inspire future work toward more practically robust solutions.
>
> ---
>
> Finally, we sincerely appreciate your deep engagement with the paper and for considering an increase in your score. If further clarification is needed on any remaining points, we would be happy to provide it.

---

> > ### Author Response · Authors · 2025-08-07
> > **Follow-up on Reviewer Feedback**
> >
> > We sincerely appreciate the time and effort you’ve dedicated to reviewing our submission and engaging in the discussion. Your feedback has been very helpful in improving the clarity and rigor of our work.
> >
> > We wanted to kindly follow up to see if there are any remaining questions or if any part of our response would benefit from further clarification. We would be more than happy to provide additional details if needed.

---

### Official Review · Reviewer_wNnw · 2025-07-03

**Clarity:** 3
**Significance:** 4
**Originality:** 3
**Rating:** 5
**Confidence:** 3

**Summary:**

This paper proposes using eight classical goodness-of-fit (GoF) tests to detect watermarks in LLM-generated text as better alternatives to common sum-based methods. It presents a systematic evaluation across three watermarking schemes, three open-source LLMs, and two datasets, and with an interesting finding that text repetition—especially prevalent in low-temperature generations—introduces detectable patterns that GoF tests can effectively exploit to enhance detection performance.

**Questions:**

I understand that GoF tests as-is don’t help directly with green-red watermarking (due to its binary pivotal statistic), so this is an open question. Given that green-red list–based watermarking is among the most widely used and dominant classes of watermarking schemes in recent work, do you see any way this paper's insights (e.g., the robustness of GoF tests) could be adapted to improve the green-red watermarking schemes?

**Ethical Concerns:**

["NO or VERY MINOR ethics concerns only"]

**Final Justification:**

The paper applies classical GoF tests to LLM watermark detection with comprehensive experiments, which I believe provide a principled and interpretable statistical framework for the domain. My main concern—the lack of practical guidance on choosing GoF tests—has been addressed with a clear plan to add concrete application scenarios discussion. Other points, such as a unified evaluation scale, were acknowledged with reasonable explanations and future directions. With the primary issue resolved and remaining points minor or forward-looking, I maintain a positive recommendation.

**Limitations:**

Yes

**Quality:**

3

**Strengths And Weaknesses:**

### Strengths
- While GoF tests are classical tools, their application to LLM watermark detection is novel and provides a principled statistical framework for understanding and improving watermarking techniques.
- The experimental setup is relatively comprehensive, covering three open-source LLMs, four temperature settings, and three types of text edits.
- The paper presents a rigorous discussion on the information-loss of sum-based statistics and the effectiveness of GoF tests in capturing CDF-level shifts, especially under low temperatures. The observation that repetition introduces useful signals is also interesting.
- The paper is generally well-structured and easy to follow.

### Weaknesses
- In each setting, only a subset of GoF tests outperforms the baseline, and no single test consistently dominates. Therefore, a more detailed discussion with practical guidance on selecting the most appropriate GoF test would greatly enhance the paper's applicability.
- While the performance gains from applying classical GoF tests for watermark detection are arguable (as mentioned above), I view this work as a benchmark-style study that systematically evaluates existing unbiased watermarking schemes using a consistent evaluation "ruler". To this end:
     * Could the authors discuss the potential for establishing a unified evaluation scale, e.g., whether the results from the eight tests can be meaningfully aggregated?
     * Including more diverse datasets if possible

---

> ### Author Rebuttal · Authors · 2025-07-29
>
> Thank you for your thoughtful review and encouraging comments. We appreciate your recognition of our contribution and your constructive suggestions for strengthening the paper. Please find our responses to your specific points below:
>
> ---
>
> ### **1. A more straightforward application of GoF tests**
>
> We appreciate your suggestion to provide more practical guidance.
>
> Indeed, **no single GoF test consistently dominates across all scenarios**—a fact we highlighted in Table 4. This diversity reflects the **classical trade-offs in hypothesis testing**: some tests are more sensitive to discrepancies in the tails (e.g., Anderson–Darling), while others emphasize central deviations (e.g., Watson’s test). Rather than prescribing a single solution, our goal in this paper is to **empirically reveal these nuanced differences** and encourage the use of GoF tests as a **flexible toolkit** for watermark detection.
>
> That said, we agree that providing more actionable recommendations is valuable. **We have summarized which GoF tests perform well under specific conditions** (e.g., low vs. high temperature, common vs. targeted edits) in Table 4. We also view **designing adaptive selection strategies**—that choose tests based on input characteristics—as an **important and promising direction** for future work.
>
> ---
>
> ### **2. Possibility of a unified evaluation scale**
>
> Thank you for this thought-provoking suggestion.
>
> While our current focus is on evaluating individual GoF tests, we agree that aggregating multiple tests into a unified scale or ensemble could further improve detection. In classical statistics, combining test statistics (e.g., via p-value aggregation or voting rules) is often used to **enhance power while maintaining Type I error control**. Applying these techniques in watermark detection, especially under editing or adversarial perturbations, is an interesting direction that could yield practical benefits.
>
> We will add a discussion in Section 5 to explicitly raise this as a potential future extension of our work.
>
> ---
>
> ### **3. More diverse datasets**
>
> We agree that evaluating on additional datasets is always beneficial and would further validate the generality of our findings.
>
> Our current selection of C4 and ELI5 **follows previous studies \[1–5]** and provides **complementary text domains**: C4 for factual completions and ELI5 for long-form generation. These allow us to capture a range of generation behaviors and model outputs.
>
> ---
>
> ### **4. How to improve the green-red watermarking scheme?**
>
> We appreciate your engagement with this limitation.
>
> As you noted, the green-red list watermark produces binary pivotal statistics (i.e., whether each token is green), which fundamentally restricts the applicability of many GoF tests. Most GoF tests are designed for continuous or finely discretized data, where empirical CDFs offer rich signal. In contrast, binary statistics yield only counts, reducing the problem to a binomial test.
>
> To make GoF-style methods applicable in this context, a promising direction is to **design enriched decoders** that produce more informative pivotal statistics beyond the binary green/red indicator. For instance, such decoders could **expose additional features** like token ranks, transition patterns between green and red tokens, or context-aware scores that reflect how strongly a token conforms to the watermarking policy. These enriched statistics would enable a more fine-grained analysis of distributional shifts and allow GoF tests to be meaningfully applied. Developing such decoders could bridge the gap between the widely adopted green-red watermarking scheme and the broader GoF-based detection framework explored in our work. However, this requires additional watermark design, which is beyond the scope of our current study.
>
> ---
>
> ### **References**
>
> [1] Li, Xiang, et al. "Robust detection of watermarks for large language models under human edits." arXiv preprint arXiv:2411.13868 (2024).
>
> [2] Kuditipudi, Rohith, et al. "Robust distortion-free watermarks for language models." arXiv preprint arXiv:2307.15593 (2023).
>
> [3] Li, Xiang, et al. "A statistical framework of watermarks for large language models: Pivot, detection efficiency and optimal rules." The Annals of Statistics 53.1 (2025): 322-351.
>
> [4] Kirchenbauer, John, et al. "A watermark for large language models." International Conference on Machine Learning. PMLR, 2023.
>
> [5] Dathathri, Sumanth, et al. "Scalable watermarking for identifying large language model outputs." Nature 634.8035 (2024): 818-823.

---

> > ### Comment · Reviewer_wNnw · 2025-08-04
> >
> > Thank you to the authors for the detailed response. In general, I still maintain a positive attitude towards this work and do not have significant concerns. Overall, the clarifications provided make sense to me.
> >
> > My only remaining comment relates to the first point: I recognize that the paper already "summarizes that GoF tests perform well under certain conditions (e.g., low vs. high temperature, common vs. targeted edits)", and I do not expect the work to propose adaptive selection strategies. However, I encourage the authors to expand this discussion by **listing concrete example scenarios**—i.e., what typical real application cases favor which tests—since in practice this is not always an easy or straightforward choice. Given that GoF tests are just classical statistical tools, adding this discussion would greatly help address potential concerns about practical usefulness.

---

> > > ### Author Response · Authors · 2025-08-05
> > >
> > > Thank you for your continued positive feedback and for clarifying your comment. We agree that discussing concrete example scenarios would enhance the practical usefulness of our work. In the next revision, we will expand this discussion to include guidance on which GoF tests may be more suitable under different real-world conditions.
> > >
> > > Specifically, we plan to add a column indicating **concrete application scenarios** for each setting (including but not limited to):
> > >
> > > - **Code generation** typically corresponds to a **low-temperature** setting
> > > - **Open-ended text generation** is often associated with a **high-temperature** setting
> > > - **Homework detection** commonly involves **common text edits**
> > > - **Internal API leakage** may involve **information-rich edits**
> > >
> > > We believe this addition will help readers better understand how to apply our findings in practice.

---

> > > > ### Comment · Reviewer_wNnw · 2025-08-05
> > > >
> > > > Thank you for outlining the concrete revision plan. I am satisfied with it and have no further concerns. I will maintain my positive score and update the minor score accordingly. Good luck!

---

> > > > > ### Author Response · Authors · 2025-08-05
> > > > >
> > > > > Thank you for your encouraging feedback and for taking the time to review our work. We sincerely appreciate your support and are glad that our revision plan addresses your concerns. Thank you again for your constructive comments and kind wishes!

---

### Author Response · Authors · 2025-08-05
**Appreciation and Welcome Further Discussions**

We want to thank all the reviewers for your dedicated efforts and valuable feedback on our manuscript. As the rebuttal deadline is approaching, we are eager to address any remaining questions that you may have. We welcome any additional comments or suggestions for clarification and are prepared to conduct further experiments if needed. We look forward to your guidance to help us improve our paper further.

---

### Note · Authors · 2025-08-11

**To the Area Chair:**

We sincerely thank you for your time and effort in overseeing the review process and we thanks all the reviewers for their thoughtful discussions and helpful suggestions. The following is our summary to Author-Reviewer discussion.

Throughout the discussion period, we addressed **nearly all concerns** raised by the reviewers and received **positive feedback** from all of them.

* **Reviewer wNnw and Reviewer jT6m**: We fully addressed their comments and provided concrete revision plans. Both reviewers expressed strong support for the paper and showed a positive and encouraging attitude toward our work.
* **Reviewer qCSQ**: The main discussion centered on our **choice of LLMs**. We clarified the rationale for using the selected models and explained why much of the watermarking community has not focused on the most recent LLMs. We also highlighted the **practical utility and low cost** of applying GoF tests. While the reviewer noted that not all low-entropy settings are equally meaningful—a point we fully agree with—we believe this does not affect the main contribution of our work: introducing GoF tests to **enrich the watermark detection toolkit**. We will include this point in our paper’s limitations. All other concerns were successfully addressed together with concrete revision plan, and the reviewer responded positively and expressed hope to see the paper presented at the conference.
* **Reviewer aMbH**: The discussion focused on the **asymptotic nature of GoF tests**. The reviewer expressed concern about the ability to tightly control FPR without a non-asymptotic distribution. In response, we provided both numerical evidence and theoretical reasoning to support that a non-asymptotic distribution is **not required** to maintain strong Type I error control. We also emphasized that GoF tests are not meant to replace sum-based or non-asymptotic methods but rather serve as **complementary tools** with different strengths. We plan to include a clear discussion in the paper outlining both the **rationale** for using GoF tests and the **potential risks** associated with their asymptotic properties. Although this reviewer has not yet responded to our most recent reply, the reviewer previously indicated a willingness to consider increasing the score, given that most concerns had been addressed.

We believe the Author-Reviewer discussion has led to meaningful improvements in both the clarity and rigor of our work.

---

### Decision · Program_Chairs · 2025-09-17

**Decision:**

Accept (spotlight)

**Comment:**

This paper presents the application of eight classical goodness-of-fit (GoF) tests to improve detection of watermarks in LLM-generated text. Rather than relying on the usual sum-based detection rules, the paper employes GoF tests in a binary hypothesis framework; comparing the empirical distribution of pivotal statistics to theoretical expectations. Evaluated across three watermarking schemes, various open-source LLMs, and datasets, GoF tests generally outperform standard approaches, especially in cases of low-temperature generations with repetitive text or under adversarial conditions. By exploiting distributional patterns rather than aggregate scores, GoF methods offer improved robustness and detection power, even when watermark integrity is compromised.

Overall, using a goodness-of-fit test for LLM watermark detection is an interesting idea. All reviewers agree to accept the paper, and I also vote for acceptance.